# Spatio-temporal variation of radionuclide dispersion from nuclear power plant accidents using FLEXPART mini-ensemble modeling

S. Omid Nabavi[1], Theodoros Christoudias[1], Yiannis Proestos[1], Christos Fountoukis[2], Huda Al-Sulaiti[2], Jos Lelieveld[1,3]

[1]Climate and Atmosphere Research Centre, the Cyprus Institute, Nicosia, 2121, Cyprus (o.nabavi@cyi.ac.cy)
[2]Qatar Environment and Energy Research Institute (QEERI), Hamad Bin Khalifa University, Doha, Qatar
[3]Atmospheric Chemistry Department, Max Planck Institute for Chemistry, Mainz, 55128, Germany

*Correspondence to*: Jos Lelieveld (jos.lelieveld@mpic.de)

**Abstract.**

We investigate the spatiotemporal distribution of radionuclides including Iodine-131 ($^{131}$I) and Cesium-137 ($^{137}$Cs), transported to Qatar from fictitious accidents at the upwind Barakah Nuclear Power Plant (B-NPP) in the United Arab Emirates (UAE). To model the dispersion of radionuclides, we use the Lagrangian particle/air parcel dispersion model FLEXible PARTicle (FLEXPART) and and FLEXPART coupled with the Weather Research and Forecasting model (FLEXPART-WRF). A four-
member mini-ensemble of meteorological inputs is used to investigate the impact of meteorological inputs on the radionuclide dispersion modelling. The mini-ensemble includes one forecast dataset (GFS) and three (re)analysis datasets (native resolution and downscaled FNL, and downscaled ERA5). Additionally, we explore the sensitivity of the radionuclide dispersion simulations to variations in the turbulence schemes, as well as the temporal and vertical emission profiles, and the location of emission sources. According to the simulated age spectrum of the Lagrangian particles, radionuclides enter southern Qatar
about 20 to 30 hours after release. Most of the radionuclide deposition in the study area occurs within 80 hours after release. The most populated areas of Qatar coincide with moderate $^{131}$I concentrations and $^{137}$Cs deposition, while uninhabited areas in southern Qatar receive the highest amounts. A larger number of long-lived particles is found in the FNL-based simulations, which is interpreted as a greater dispersion of particles at a greater distance from the emission location. The highest simulated $^{131}$I and $^{137}$Cs deposition shows a pronounced spatiotemporal pattern. The largest impacts are found in the south/southeast of
Qatar, during the early-daytime development of the boundary layer, and during the cold period of the year. The results show remarkable differences in the spatiotemporal distribution of $^{131}$I and $^{137}$Cs simulations based on the FNL and GFS datasets, which share a common base meteorological model. As part of a sensitivity analysis involving different model setups, changing the emission point from B-NPP to Bushehr-NPP (Bu-NPP) results in a reduced transfer of radioactive materials to Qatar, except in the spring season. Bu-NPP simulations reveal distinct spatial patterns, with peak $^{131}$I concentrations and $^{137}$Cs
deposition observed in the northern and eastern Qatar during winter and spring.

# 1 Introduction

Modeling the spatiotemporal distribution of radioactive compounds using chemical transport models (CTMs), especially after large nuclear accidents, has received widespread attention (Chino et al., 2011, Stohl et al., 2012, Christoudias and Lelieveld, 2013, Evangeliou et al., 2017). Whether explicitly stated or not, these studies aim to determine the extent and transport of radionuclides at various spatial scales. They also aim to improve the performance of CTMs, which are at the core of preparedness programs for potential nuclear accidents (or releases) associated with the growing number of nuclear facilities (Farid et al., 2017). However, case studies of large accidents, which have a typical duration of a few days to a few weeks, are not suitable for studying the effects of diurnal and seasonal (atmospheric) changes on radionuclide dispersion, and the sensitivity to the meteorological (re-)analyses driving the simulations.

Maurer et al. (2018) found that the performance of dispersion models strongly depends on the successful modeling of boundary layer processes, which vary with season and time of day. Meteorological inputs to the CTMs are generated by atmospheric models, which may exhibit variability when simulating atmospheric conditions over nuclear accident areas (Arnold et al., 2015). Long et al. (2019) studied the effects of the East Asian northeast monsoon on the transport of radionuclides from the Fukushima nuclear power plant accident to the tropical western Pacific and Southeast Asia. They found that in these regions, radioactivity levels are lower than in other regions of the Northern Hemisphere, which is due to the late arrival of the radionuclide plumes carried by the monsoon circulations. That is, the dispersion of radionuclides from this accident could potentially be different under other atmospheric conditions, which are only captured by the hypothetical, iterative simulation of this event at different times of the day and year.

While the licensing of new nuclear power plants is rare in the Western world and many older Soviet-era stations are approaching decommissioning, the Middle East/North Africa (MENA) region has seen a rise in the planning, proposal, and recent operation of nuclear facilities. The Barakah Nuclear Power Plant (B-NPP) is the latest example, following the Bushehr NPP (Bu-NPP), to become operational in a region with unique climatological conditions. Surprisingly, there is limited literature addressing the potential risks of radionuclide dispersion in this region, unlike the extensive research conducted in Europe, Japan, and the USA.

The main objective of this study is to investigate the spatio-temporal variability of radionuclide transport, surface concentration, and deposition after potential nuclear accidents, using a mini-ensemble of meteorological inputs. Gaseous and aerosol radionuclide tracers, Iodine-131 ($^{131}$I) and cesium-137 ($^{137}$Cs), are assumed to be released by fictitious accidents at the B-NPP in the United Arab Emirates (UAE). The amount of $^{137}$Cs that is deposited on the surface is important due to its relatively long half-life (about 30 years), whereas the $^{131}$I surface-level concentration (about 8 days half-life) is important for human health and the biosphere in the short term (Tsuruta et al., 2019, Pisso et al., 2019, Takagi et al., 2020, Kinase et al., 2020, Wai et al., 2020).

For the dispersion modeling we use the Lagrangian particle model FLEXible PARTicle (FLEXPART) (Stohl et al., 1998, Stohl et al., 2005, Brioude et al., 2013). Lagrangian particle dispersion models (LPDMs) model the trajectory of each particle (Nabi

et al., 2015) instead of the transport of air pollutants by numerically solving the equation for the conservation of mass (Moussiopoulos, 1997, Zhang and Chen, 2007) notably done with Eulerian CTMs (Brioude et al., 2013). LPDMs are computationally more efficient as they calculate advection and diffusion only for the location of each particle and not for the entire model domain. However, LPDMs also have limitations. They suffer from numerical errors in the interpolation of meteorological fields in space and time. In some cases, the particles may not remain well-mixed during simulation (Brioude et al., 2013). This is mainly due to the treatment of the stochastic motion of the particles and/or the mass balance of vertical velocity with the horizontal winds. Regardless of the formalism used, Girard et al. (2016) have shown that uncertainties in meteorological fields, namely wind speed and direction and precipitation, and emission rate can contribute significantly to errors in the dispersion modeling of radionuclides. According to Galmarini et al. (2004), there are three common methods for sensitivity analysis of LPDM simulations with respect to meteorological fields: i) generating perturbations in the horizontal and vertical position of particles, ii) using a single meteorological model with perturbations in initial conditions and/or model physics, and iii) using a number of different meteorological models. By adopting the third approach, we use a four-member mini-ensemble in which the performance of the forecast member, forced by 6-hourly data from the Global Forecast System (GFS), is compared to the members based on (re)analysis datasets. (Re)analysis-based simulations (unavailable in a real-world scenario) are expected to be closer to actual values than forecast-based ones (Leadbetter et al., 2022). Two members of the mini-ensemble are forced by dynamically downscaled meteorological inputs to investigate the effects of downscaling to higher resolution in dispersion modeling of radionuclides. Moreover, we investigated the sensitivity of our simulations to different turbulence schemes under convective conditions, as well as variations in the vertical profile, temporal profile, and point of emission. Specifically, we examined two different vertical profiles, two points of emission, and two temporal profiles of emissions, with one main variant and one secondary variant for each variation. The main variants were used in most of the manuscript, while the secondary variants were used in a small sensitivity test to evaluate the impact of these variations on our results.

Finally, to study the potential risks to the local population, the radionuclide simulations are examined in relation to the population density of the catchment region of interest (Qatar). To our knowledge, this is the first study to investigate potential radionuclide releases in the region, and our results could provide important information for developing risk assessment and preparedness plans. Specifically, our findings, which account for variations in meteorological conditions and model inputs, could help identify areas and times of the year that are more susceptible to higher concentrations of radionuclides, and thus aid in developing appropriate mitigation strategies. The paper is organized as follows: section 2 describes the LPDM, meteorological inputs, study area, and source term of radionuclides used in this study. The results are presented and discussed in section 3. The conclusions of the study are presented in section 4.

## 2 Mini-ensemble model configuration

A brief description of the FLEXPART modeling structure follows.

## 2.1 FLEXPART and FLEXPART-WRF dispersion modeling

We refer the reader to Stohl et al. (2005), (Pisso et al., 2019), and Brioude et al. (2013) for detailed descriptions of the FLEXPART and FLEXPART-WRF models. Here, only the principles of FLEXPART Lagrangian modeling are discussed to facilitate the presentation of the results in the next section. FLEXPART was developed as a LPDM based on the zero acceleration scheme (Eq. 1). It solves a Langevin equation (Eq. 2) to model the trajectories of Lagrangian particles. The new position of the particles is influenced by large-scale winds, local turbulence (stochastic component), and mesoscale motions.

$$X(t + \Delta t) = X(t) + v(X, t)\Delta t \ (1)$$

$$\frac{dX}{dt} = v[X(t)]$$

where $t$ is time, $\Delta t$ is the time increment, X is the position vector, and $v = \bar{v} + v_t + v_m$ is the wind vector composed of the grid-scale wind $\bar{v}$, the turbulent wind fluctuations $v_t$, and the mesoscale wind fluctuations $v_m$. FLEXPART also quantifies changes in the mass, or mixing ratio, of transported particles carried away by calculating various removal processes (Stohl et al., 2005, Grythe et al., 2017). Turbulent motions $v_{t_i}$ for wind components $i$ are parameterized assuming a Markov process based on the Langevin equation (Eq. 2).

$$dv_{t_i} = \alpha_i(x, v_t, t)dt + b_{ij}(x, v_t, t)dW_j \ (2)$$

where $\alpha$ is the drift term, $b$ is the diffusion term, and $dW_j$ are incremental components of a Wiener process with mean zero and variance $dt$, that are uncorrelated in time (Stohl et al., 2010). The minimum value of time step $\Delta t_i$ is 1 second. $\Delta t_i$ is used only for the horizontal turbulent wind components of Equation 3.

$$\Delta t_i = \frac{1}{ctl}min(\Delta \tau_{L_\omega}, \frac{h}{2\omega}, \frac{0.5}{\frac{\partial \sigma_\omega}{\partial z}}) \ (3)$$

In Equation 3, $h$ represents the height of the atmospheric boundary layer and $z$ denotes the height of the model level. The constant $ctl$ represents a predefined characteristic time scale. $\tau_{L_\omega}$ is the Lagrangian timescale for the vertical velocity autocorrelation. Additionally, $\omega$ represents the turbulent vertical wind component, while $\sigma_\omega$ represents its standard deviation. To solve the Langevin equation for the vertical wind component, a shorter time step $\Delta t_\omega = \frac{\Delta t_i}{ifine}$ is used. Under convective conditions, when the turbulence is skewed, larger areas are occupied by downdrafts (instead of updrafts). This can lead to higher surface concentrations (deposition) in areas near pollution sources (Pisso et al., 2019). To investigate the sensitivity of radionuclide quantities to the turbulence scheme used, one sensitivity run by FLEXPART-WRF is performed after replacing the standard Gaussian turbulence model (GTM) with the skewed turbulence model (STM) (Luhar et al., 1996, Cassiani et al., 2015). The results of either scheme are used to calculate the vertical velocity component of the drift term in Eq. 2. The implementation of STM requires shorter time steps, $dt$ in Eq. 2, to better resolve turbulence in the convective planetary boundary layer. Therefore, we use $ctl$=10 and $ifine = 10$ as recommended by Brioude et al. (2013) and Pisso et al. (2019),

in this sensitivity run. For the computation of $\sigma_{v_i}$ and $\Delta\tau_{L_i}$, FLEXPART uses the parameterization scheme proposed by Hanna (1982). To account for mesoscale motions, a method similar to that of Maryon (1998) is used.

In radionuclide dispersion modeling, particle mass reduction of occurs primarily through three processes: radioactive decay, dry deposition, and wet deposition. The following exponential equations characterize radioactive decay (Equation 4) and wet deposition (Equation 5).

$$m(t + \Delta t) = m(t)exp(-\Delta t/\beta) \quad (4)$$

where $m$ is the particle mass and the time constant $\beta = \frac{T_{\frac{1}{2}}}{\ln(2)}$ is determined from the radionuclide half-life $T_{\frac{1}{2}}$.

$$m(t + \Delta t) = m(t)exp(-\Lambda\Delta t) \quad (5)$$

The scavenging coefficient $\Lambda$ is calculated differently depending on whether the particles are in the aerosol or gas phase and whether the scavenging takes place inside or below the clouds (Stohl et al., 2005).

In this study, we use FLEXPART 10.4 and a modified version of FLEXPART 9.0.2 to input meteorological simulations from the Weather Research and Forecasting (WRF) model (Grell et al., 2005) (hereafter referred to as FLEXPART and FLEXPART-WRF, respectively). Compared to previous versions, FLEXPART has undergone a significant revision of the wet deposition calculation. The latest updates incorporate the dependence of wet deposition on aerosol particle size and precipitation type (Grythe et al., 2017). FLEXPART calculates wet deposition only for cloudy grid cells where the precipitation rate exceeds 0.01 mm h$^{-1}$. Therefore, the accuracy of cloud pixel detection plays a critical role in the accuracy of the location and amount of wet deposition simulated by FLEXPART. In previous versions, including version 9.0.2 used in the development of FLEXPART-WRF 3.2, in-cloud grid cells are defined as those with precipitation and relative humidity above 90% (Brioude et al., 2013). The grid cells below the in-cloud grid cells up to the surface are defined as below-cloud grid cells (Seibert and Arnold, 2013, Pisso et al., 2019). In recent updates to the FLEXPART's source code, the above threshold has been modified using the 3D cloud water mixing ratio ($q_c$) fields. The threshold of $q_c > 0$ ($q_c = 0$) now identifies grid cells within the cloud (below the cloud) (Pisso et al., 2019). Therefore, the differences in approaches in calculating scavenging between FLEXPART and FLEXPART-WRF may partially explain discrepancies between simulations for the wet season. Pisso et al. (2019) have discussed this and other updates in more detail in the latest version of FLEXPART. For dry deposition, for all particles below twice the reference height ($h_{ref}$), the reduction in particle mass is calculated using Eq. 6:

$$\Delta m(t) = m(t)[1 - exp\left(\frac{-v_d(h_{ref})\Delta t}{2h_{ref}}\right)] \quad (6)$$

$v_d$ is the dry deposition velocity calculated as the ratio of $v_d(z) = \frac{F_C}{C(z)}$ for $h_{ref}$ of 15 m. $F_C$ and $C(z)$ are the flux and concentration of a species at height z within the layer at constant flux. If the necessary information to parameterize of dry deposition of gases and particles is not available, $v_d$ can be assumed to be constant. The concentration of a given gas or aerosol species in a grid cell is equal to the weighted average of the total particle mass within the grid cell divided by the volume of the grid cell (Pisso et al., 2019), as defined in Equation 7.

$$C(z) = \frac{1}{v}\sum_i^N (m_i f_i) \ (7)$$

where V is the volume of the grid cell, $m_i$ is the particle mass, $N$ is the total number of particles, and $f_i$ is the fraction (weight) of the mass of particle $i$ assigned to the particular grid cell. The amount of dry and wet deposition over the given grid cell is accumulated by default over the time dimension of the output, unless the age composition of the air parcels is required, as is the case in this study. A unique feature of FLEXPART/FLEXPART-WRF is the grouping of simulations based on the age of the Lagrangian particles. This means that the number of concentrations and deposition at each time step is obtained from aggregating the simulations over the additional dimension of particle age. In this study, we use the age spectrum of Lagrangian particles to investigate the age composition of radioactive materials from the source to the receptors. Air parcel ages are studied at an hourly resolution. As a result, the output grid, which has a horizontal resolution of 10 km and 14 vertical levels from 5 to 5000 m agl, gains two dimensions time and age spectra with a same length of 96 (hours). Thickness-weighted averages of simulated concentrations near the surface, hereafter referred to as near-surface concentrations, are calculated from concentrations within the bottom four model layers between 5 and 100 m agl (with layer thicknesses of 5 m, 5 m, 40 m, and 50 m). To statistically compare the age distributions of the mini-ensemble members, we use the maximum normalized difference according to Equation 8. Assuming that a and b are two air parcel age distributions that have been smoothed with the Gaussian kernel (Chung, 2020), their maximum normalized difference is calculated as the maximum value of the absolute differences between a and b divided by the maximum value of a and b. Larger variations in the distributions are indicated by higher values of this indicator (Jin and Kozhevnikov, 2011).

$$\text{maximum normalized difference } = \frac{\max(\text{abs}(a-b))}{\max(\max(a),\max(b))} \ (8)$$

## 2.2 Meteorological data

FLEXPART and FLEXPART-WRF are driven offline by gridded meteorological fields from numerical weather prediction models. In this study, we obtain meteorological products from the GFS (NCEP, 2015a), the NCEP final analysis (FNL) (NCEP, 2015b), and the ECMWF reanalysis fifth generation (ERA5) (Hersbach et al., 2020). GFS is a fully coupled model that represents the interactions between oceans, land, and atmosphere. The GFS dataset, available since 2015, provides 6-hourly forecast inputs at 03, 09, 15, and 21 hours with a spatial resolution of 0.25 degrees. The FNL dataset is produced using the same base meteorological model that produces the GFS dataset. The former provides the three-hourly combination of analysis (at 00, 06, 12, and 18) and forecast meteorological fields (at 03, 09, 15, and 21) at a spatial resolution of 0.25 degrees starting in 2015. ERA5 is the latest generation of ECMWF reanalysis data, covering the period from January 1, 1950 to the present. It is generated in hourly time steps with a spatial resolution of about 0.25 degrees. GFS and FNL are used to force FLEXPART directly. In addition, WRF 4.2 is applied to dynamically downscale the ERA5 and FNL to be used as the input to FLEXPART. They are used in FLEXPART-WRF with higher downscaled spatio-temporal resolutions of 10 km and hourly. Though we devoted efforts to incorporate ERA5 reanalysis data directly as inputs for FLEXPART, we were hindered by technical

difficulties encountered with the data preprocessor tool, flex_extract, thereby impeding our ability to do so. While we were working to resolve the technical issues with the developers, a solution was not readily available during the writing and revision of this manuscript. We have indirectly employed ERA5 reanalysis data from the Research Data Archive (RDA) by incorporating it into the WRF model and subsequently into the FLEXPART-WRF. This approach has enabled us to assess the influence of ERA5 on our downscaled simulations and to compare it with FNL. To the best of our knowledge, this is the first time that the ERA5 reanalysis has been used with the FLEXPART-WRF model setup. Table 1 provides an overview of the WRF model configuration.

**Table 1 Overview of the WRF model configuration.**

| | |
|---|---|
| **Dynamics** | Non-hydrostatic |
| **Initial and boundary condition data** | FNL/ERA5 |
| **Temporal interval of boundary data** | 3 h/1 h |
| **Resolution** | 10 km x 10 km |
| **Extent of domain** | 17°N-33°N and 40°E-60°E |
| **grid-nudging** | On |
| **PBL Scheme** | YSU |
| **Cumulus parameterization** | Grell 3D ensemble scheme |
| **Surface layer parameterization** | Noah land surface scheme |
| **Terrain and land use data** | USGS |

Table 2 provides an overview of the input data and the corresponding simulation codes. For comparison of the above meteorological inputs with observations, daily total precipitation and daily averages of wind speed and temperature data are obtained from 157 climate stations within the model domain. These observations are freely available from the Global Surface Summary of the Day (GSOD) database which is maintained by the National Oceanic and Atmospheric Administration (NOAA). The GSOD data is subject to a rigorous two-tier quality control (QC) process (https://www.ncei.noaa.gov/data/global-summary-of-the-day/doc/readme.txt, last accessed: May 5, 2023). The first level entails the application of meticulous automated QC procedures that effectively purge the raw data of random errors. The second level of QC is performed to create daily summaries. However, there is still a slight likelihood of unknown errors being present within the GSOD dataset.

**Table 2 Summary of meteorological inputs used to run FLEXPART and FLEXPART-WRF.**

| Inputs | Simulation code | Spatial resolution | Temporal resolution | Time coverage | Downscaled | Type of input | Wet deposition scheme |
|---|---|---|---|---|---|---|---|
| | | | | | | | |

| GFS | FLEXPART | 0.25 degrees | 6-hourly | 2011-present | - | forecast | Grythe et al. (2017) |
|---|---|---|---|---|---|---|---|
| FNL | FLEXPART/FLEXPART-WRF | 0.25 degrees/ 10 km | 3-hourly/ hourly | 2015-present | -/✓ | analysis | Grythe et al. (2017)/Seibert and Arnold (2013) |
| ERA5 | FLEXPART-WRF | 10 km | hourly | 1950-present | ✓ | reanalysis | Seibert and Arnold (2013) |

**2.3 Emission scenario and study area**

We simulate fictitious nuclear accidents at the B-NPP (Fig. 1-A) in which 6.9 kg (22 PBq) of $^{137}$Cs and 0.042 kg (192 PBq) of $^{131}$I are released during the first 24 h of each 96-hour simulation period. These amounts are the upper bounds of emissions
estimated by Babukhina et al. (2016) for the Fukushima Daiichi nuclear accident in March 2011. Although we simulate a fictitious release of radioactivity at a level comparable to the Fukushima nuclear accident, our study does not intend to replicate past accidents or simulate a specific real-world case. These source terms are used to provide a real-world comparison that gives the reader a tangible point of reference.

The number of Lagrangian particles required is dictated by the specific problem at hand (Papagiannopoulos et al., 2020,
Thompson et al., 2015, Fast and Easter, 2006). The computational time scales increase linearly with the increasing number of particles, while the statistical error of the simulations decreases with the square root of the particle density (Pisso et al., 2019). In this study, a total number of $10^4$ Lagrangian particles are released uniformly during the 24-hour emission period. To assess if this rate is sufficient, we study the dry deposition process, which is directly influenced by the boundary layer conditions. We performed a preliminary test run with GFS data with a 10-fold increase in particles. As shown in S1, the simulated dry
deposition of $^{137}$Cs and $^{131}$I and the wet deposition of $^{137}$Cs do not undergo significant changes with the increase of an order of magnitude over Qatar.

Regarding the height of the releases, the particles are emitted at model levels between 100 and 300 m above ground level (agl) above the emission point. Each simulation starts at the beginning of each day and lasts 96 hours (Fig. 1-B). We run this scenario for every day of the year 2019, resulting in a total of 1460 simulation days (365 days of the year x 4 forward simulation days).
In the following analysis, when reporting particle ages associated with the simulations, we always refer to the time when a particle is released.

As expected, the temporal and spatial characteristics of the release of radioactive materials (like other pollutants) significantly impact the simulations. In this study, we perform three sensitivity analyses on the release conditions, in addition to one sensitivity test on the skewed turbulence model (STM), using FLEXPART-WRF based on ERA5 reanalysis. Specifically, we

conduct the Continuous Release Sensitivity Test (CRST), where pollutants are released continuously for 96 hours, as opposed to only the first 24 hours of the accident. Following Morino et al. (2011), we also increase the upper limit of release height from 300 to 700 meters (referred to as the Release Height Sensitivity Test (RHST)). The third sensitivity test, designated as the Release Location Sensitivity Test (RLST), is conducted by releasing radionuclides from the Bushehr NPP (Bu-NPP) and investigating the impact of source-receptor position on the transport of radionuclides, particularly in relation to seasonal variations in atmospheric patterns. The simulation domain extends between 17°N-33°N and 40°E-60°E, and for post-processing we analyse simulations over Qatar, between 24.25°N-26.35°N and 50.65°E-51.75°E.

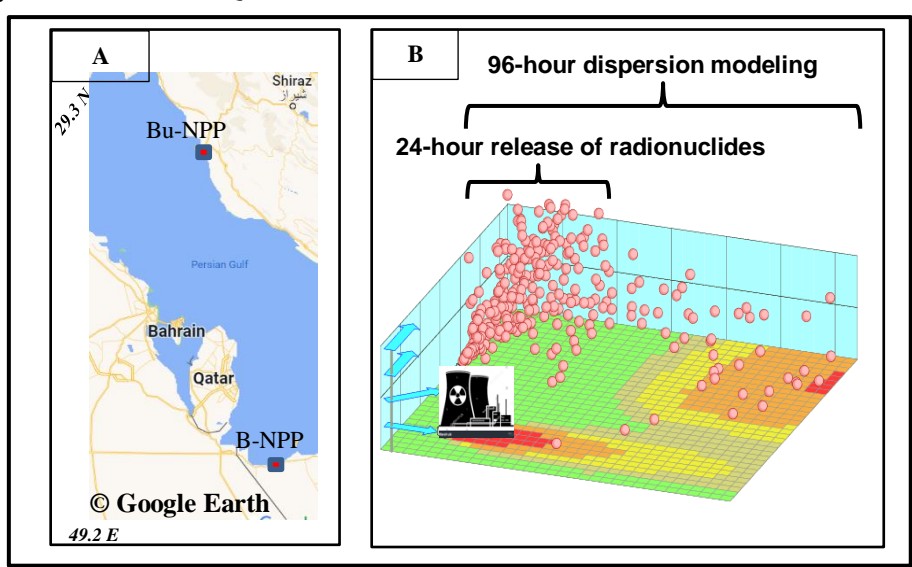

**Figure 1 A: The study area includes the Barakah Nuclear Power Plant (B-NPP) and the Bushehr New Nuclear Power Plant (Bu-NPP), both represented by red squares, as well as the country of Qatar. The base map and overlay information, excluding the names of the NPP facilities, are sourced from Google Earth. The LPDM simulation cycle is schematically depicted in Figure B, which is originally retrieved from https://www.janicke.de/en/lasat.html (last accessed: May 5, 2023).**

## 3 Results and discussion

### 3.1 Meteorological inputs vs. observations

We compare the relevant meteorological input fields including daily total precipitation, daily mean surface wind speed and temperature with observations from 157 stations spanning the model domain (Table 3). A relatively large discrepancy between observations and model inputs (as well as among different model inputs, as depicted in S2 and S3) is attained for precipitation. The average Spearman correlation coefficient (r) of ~0.41 and root mean square error (RMSE) of 3.2 mm indicate a moderate relationship between simulated precipitation inputs and observations in all members. The ERA5-WRF and FNL datasets have the highest correlation (0.44) and lowest RMSE (3 mm) compared to observations, respectively. All datasets, especially the downscaled ones, underestimate precipitation amounts, resulting in an average mean bias error (MBE) of -0.22. The seasonal

boxplots of precipitation datasets (S4-A) reveal that the underestimation of precipitation is most pronounced in extreme precipitation events, particularly in the downscaled simulations. Specifically, the upper quartile of precipitation in observations during the rainy seasons of winter and fall are approximately 7.5 mm and 5.5 mm, respectively. In contrast, the corresponding values in the FNL, GFS, ERA5-WRF, and FNL-WRF simulations are markedly lower, amounting to approximately 6 mm and 3.1 mm, 5.8 mm and 3.3 mm, 4.8 mm and 2.3 mm, and 2 mm and 1 mm, respectively.

The underestimation of precipitation in atmospheric modeling datasets is a well-known challenge, attributed to the complex interplay of sub-grid-scale processes that are often parameterized in the models. These parameterizations rely on assumptions and approximations, introducing uncertainties in precipitation simulations. Additionally, the intricate topography of the study area can significantly influence precipitation patterns (Tapiador et al., 2019).

The more severe underestimation observed in downscaled simulations can be partly attributed to the greater sensitivity of these simulations to the choice of physical parameterizations. The use of higher spatial resolution in downscaled simulations means that physical parameterizations are applied over smaller areas, which can magnify the impact of uncertainties in these parameterizations. For instance, parameterizations of cloud microphysics and convection can be particularly important in determining the intensity and duration of extreme precipitation events (Kain, 2004). While this study did not specifically investigate model resolutions finer than 10 km, increasing the model resolution to 1 km or less provides the capability to resolve finer subgrid phenomena. This allows for the deactivation of convective schemes designed for coarser resolutions, as the resolved deep convection becomes more prominent.

All meteorological datasets are closer to the observations when evaluating the simulations of wind speed and temperature. The ERA5-WRF dataset shows the best agreement with the wind speed observations in terms of correlation (0.65) and RMSE (1.5 m/s), but overestimates wind speed (MBE > 0.5) to a greater extent than the GFS and FNL datasets (MBE < -0.13). Based on the boxplot of wind speeds shown in S4-B, the GFS and FNL datasets underestimate wind speed mainly in low intensities, while the ERA5-WRF and FNL-WRF overestimate wind speed in high intensities. All datasets are significantly correlated with the temperature observations and with each other, as shown by the distribution of data points along the identity line (S2 and S3). The distributions of the simulated and modelled temperature exhibit the least dissimilarity (S4-C) in comparison to precipitation and wind speed. In order to deepen our understanding of the model performance at the primary emission source and receptors in Qatar, we conducted a similar analysis on 12 stations within the area encompassing the B-NPP and the state of Qatar (S5, S6, and S7). Although some variations are noted, such as an improvement in the correlations between simulated and observed wind speed, the overall patterns of performance for the input datasets over this region remain consistent with those identified earlier. Based on these findings, we infer that the FNL- and ERA5-WRF (downscaled) datasets exhibit a better correlation with observations. However, this comes at the expense of increased bias values when compared to the FNL and GFS datasets.

The inter-comparison of the input datasets shows that the FNL and GFS simulations have the best agreement, as can be expected since they are produced using the same data assimilation and forecasting system. The better agreement found between

the ERA5-WRF and FNL-WRF datasets than between the latter and the FNL dataset shows the impact of the downscaling on

the homogeneity of the meteorological inputs. It should be noted that a systematic comparison of meteorological datasets with a representative sample of surface observations is needed to determine the optimal choice of meteorological inputs for forcing a CTM for different regions. Here we have access to daily surface meteorological data, while transport modeling is performed in several layers of the atmosphere at hourly timesteps or shorter.

**Table 3: Comparison of modelled and observed daily precipitation, wind speed, and temperature.**

| Variable | Mini-ensemble members | Spearman correlation coefficient | RMSE | MBE |
|---|---|---|---|---|
| Precipitation (mm) | GFS | 0.42 | 3.1 | **-0.18** |
| | FNL | 0.42 | **3** | **-0.18** |
| | FNL-WRF | 0.37 | 3.4 | -0.27 |
| | ERA5-WRF | **0.44** | 3.3 | -0.24 |
| mean | | 0.41 | 3.2 | -0.22 |
| Wind Speed (m/s) | GFS | 0.58 | 1.7 | **-0.13** |
| | FNL | 0.58 | 1.7 | -0.15 |
| | FNL-WRF | 0.64 | 4.1 | 0.62 |
| | ERA5-WRF | **0.65** | **1.5** | 0.51 |
| mean | | 0.61 | 2.2 | 0.21 |
| Temperature (k) | GFS | 0.97 | 2.1 | **0.02** |
| | FNL | **0.98** | 2 | **0.02** |
| | FNL-WRF | **0.98** | **1.8** | 0.07 |
| | ERA5-WRF | 0.97 | 2 | -0.33 |
| mean | | 0.975 | 1.975 | -0.055 |

**3.2 Age composition of radionuclide plumes**

The seasonal distributions of the air parcel ages for [131]I and [137]Cs concentrations are shown in Figures 2 and S8, respectively. The high degree of similarity between the age distributions of [131]I and [137]Cs is due to the fact that the removal process in LPDMs only affects mass concentrations and not particle positions. Therefore, the following discussion of the ages of the [131]I particles is also valid for the ages of the [137]Cs particles. All mini-ensemble members, in particular GFS and FNL, simulate

predominantly air parcels with lifetimes of less than 20 to 30 hours. Other than fall, the age distribution of the particles is relatively similar in all other seasons. Seasonal atmospheric patterns with a low frequency of occurrence in the GFS and FNL datasets, discussed in subsection 3.3, seems to be responsible for the difference in particle age distributions in fall. Compared

to the other mini-ensemble members, the FNL-based simulations show a delayed appearance of the second peak in air particle ages. This could be due to a decrease in transport speed to the same receptors and/or a higher number of air parcels reaching areas away from the source (discussed later).

To understand the relationship between radionuclide concentrations and air parcel ages, the latter are divided into three groups corresponding to low (the top row in S9), moderate (the bottom row in S9), and high (the bottom row in Fig. 2) near-surface concentrations of $^{131}$I (see S8 and S10 for $^{137}$Cs). These categories are determined for each member based on concentrations below the 33$^{rd}$ percentile, between the 33$^{rd}$ and 66$^{th}$ percentiles, and above the 66$^{th}$ percentile, respectively. The rapid transport of dense radionuclide clouds within the region of interest is demonstrated by the strongly positively skewed age distributions of particles corresponding to moderate and high radionuclide concentrations in all members and all seasons. The inconsistency of age distributions for high and moderate concentrations in summer can be attributed to an unusual transport pattern in this season.

It is worth noting that all members of the mini-ensemble show closer agreement for air parcel ages corresponding to the moderate and high levels of $^{131}$I and $^{137}$Cs concentrations than for those corresponding to low concentrations. This is consistent with the principles of FLEXPART dispersion modeling. According to Pisso et al. (2019), the error rate of the simulations decreases with the square root of the particle density. As a result, all four members simulate the age of Lagrangian particles corresponding to moderate and high concentrations in better agreement and likely more accurately near the source. Using the maximum normalized distance, a quantitative comparison of the particle age distributions, corresponding to all concentrations, is shown in Figure 3. The age distribution in the ERA5-WRF- and FNL-WRF-based simulations shows a greater similarity than to that of the FNL-based simulations in all seasons, except fall. This is due to the dynamic downscaled meteorological inputs with the same spatio-temporal resolution and to a common simulation code in ERA5- and FNL-WRF-based simulations. The age distributions based on the FNL and GFS inputs, which are generated by a similar base model, do not show the same degree of similarity in all seasons. They have the smallest difference in fall and spring and a larger difference in winter and summer.

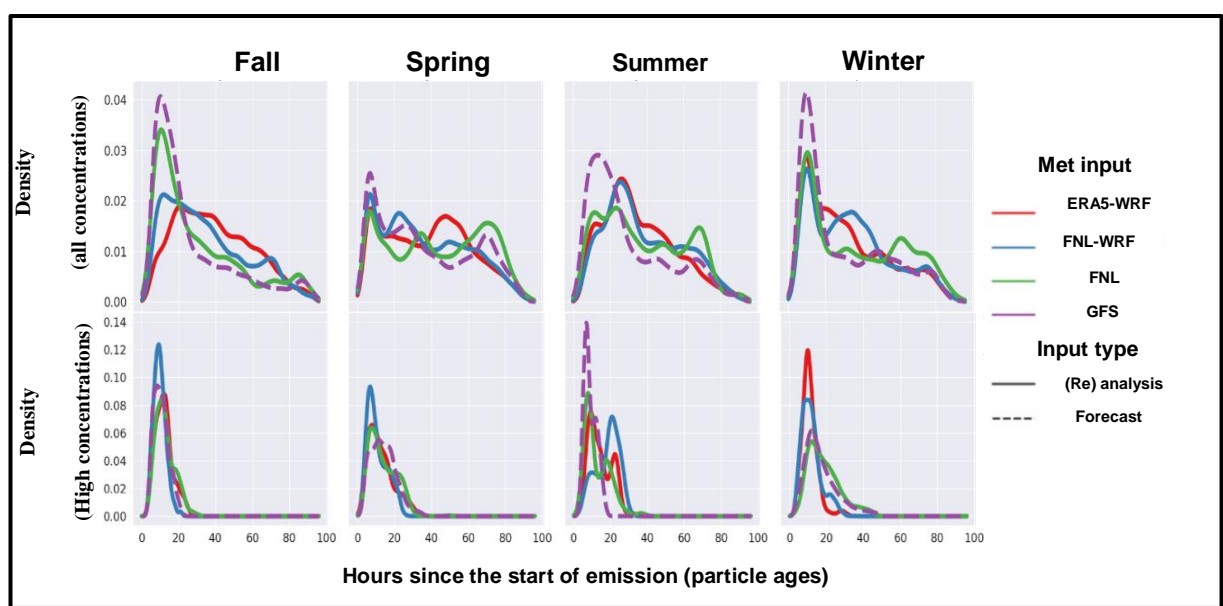

**Figure 2 A: Distributions of the air parcels corresponding to all near-surface $^{131}$I concentrations (top row) and of those above the 66$^{th}$ percentile (bottom row). y-and x-axes show density values and air parcel ages, respectively.**

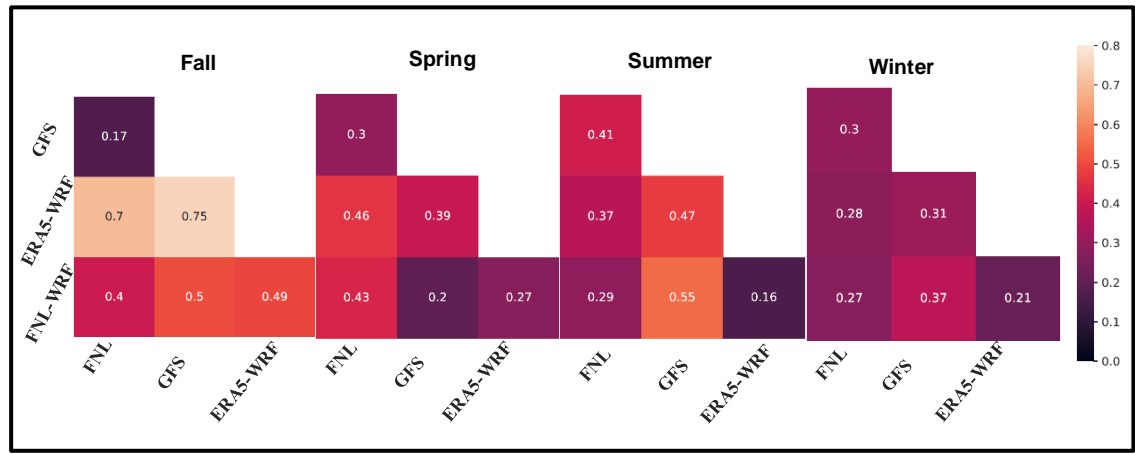

**Figure 3 Seasonal maximum normalized difference of air parcel age distributions.**

Due to the relatively long half-life of $^{137}$Cs, its deposition rate in the affected areas is of great importance. To analyze the relationship between the age composition of air parcels and the amount of $^{137}$Cs deposition, the deposition values cumulatively aggregated across time steps (j) and age spectra (i) are normalized to the total amount of $^{137}$Cs deposition in each grid cell (k) at the end of each simulation run (l).

$$^{137}\text{Cs}_{kln a_{norm\_depso}} = \begin{cases} \dfrac{^{137}\text{Cs}_{kl(n-1)n}}{\sum_{j=1}^{96}\sum_{i=1}^{95} {}^{137}\text{Cs}_{klij}} & if\ n = 2 \\[4mm] \dfrac{\sum_{j=1}^{n-1}\sum_{i=1}^{j-1} {}^{137}\text{Cs}_{klij} + \sum_{i=1}^{a} {}^{137}\text{Cs}_{klin}}{\sum_{j=1}^{96}\sum_{i=1}^{95} {}^{137}\text{Cs}_{klij}} & if\ n > 2 \end{cases} \quad (9)$$

where n is the time step with a maximum of 96 (the last time step) and $a$ is the given particle age with a maximum of n-1. $^{137}\text{Cs}_{klij} = 0$ in two conditions (1) $n >= 26$ if $i \in [1, n-25]$ and (2) $i >= j$.

Figure 4 shows the normalized deposition amounts ($^{137}\text{Cs}_{kln a_{norm\_depso}}$) in winter, when both dry and wet deposition occur in the study area. Similar deposition patterns are obtained for other seasons (S11). As shown in S12, the main reason for the small difference between the seasonal deposition patterns is the lack of precipitation and subsequent wet deposition in the region. Although the spatial pattern of the deposition varies considerably, as indicated by the range of quartiles, the median of the normalized deposition shows that about 80 percent of the deposition occurs within 80 hours after an accident. The

cumulative deposition at the end of the simulation period is mostly in the areas farthest from the source, and the total deposition reaches 100% as it approaches the end of the 96-hour simulation period. This is evident in S13, where the ages of the deposited particles peak around 20-30 hours, with a rather small number after 80 hours.

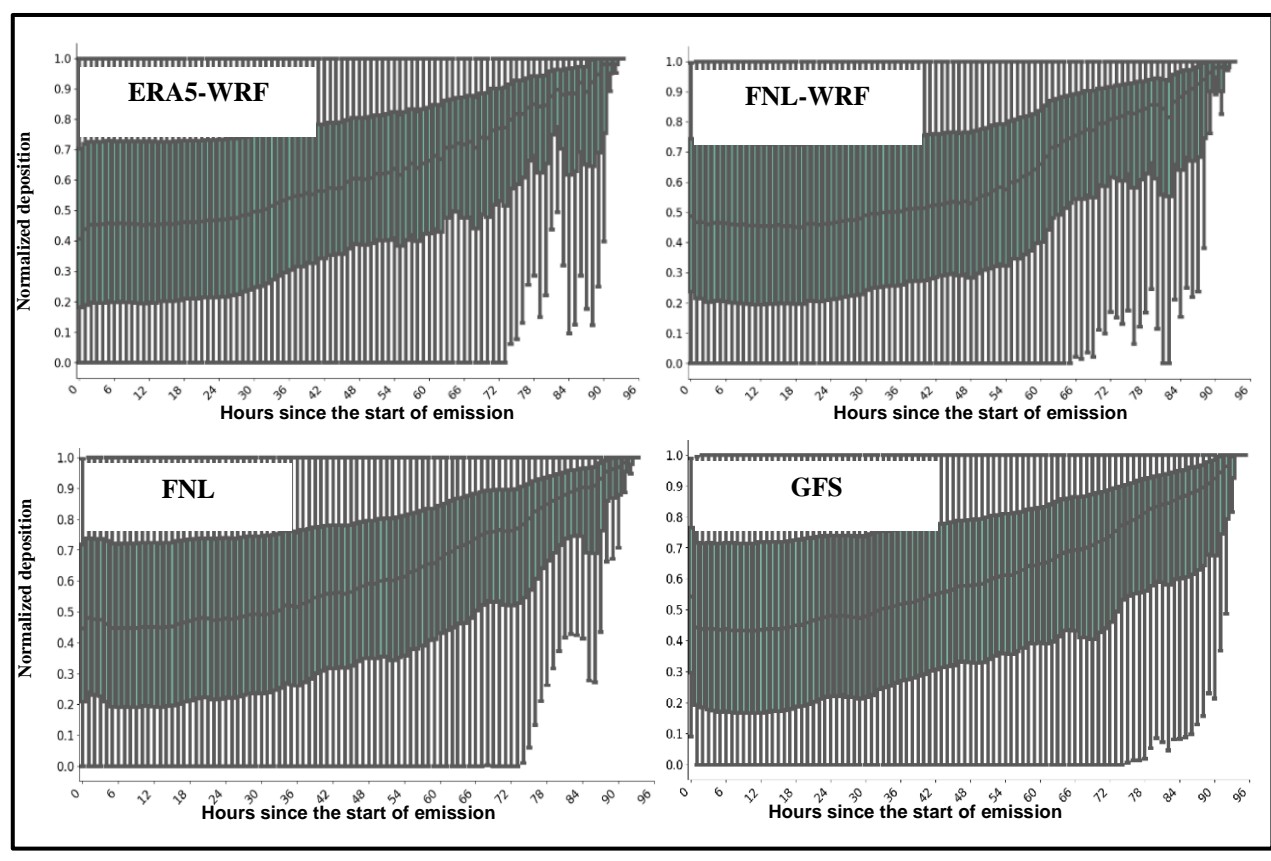

**Figure 4 The normalized cumulative $^{137}$Cs deposition to the total $^{137}$Cs deposition of each 96-hour simulation period at each grid cell in winter. The x-axis shows the age of the Lagrangian particles and the y-axis is the normalized deposition. The boxes show the quartiles of the normalized deposition. The whiskers represent the range between 1.5 times the interquartile range above the upper quartile and below the lower quartile.**

## 3.3 Spatio-temporal distribution of radionuclides

In this section, we explore the seasonal and diurnal changes in the concentrations and deposition of radionuclides within the study area. To analyze the radionuclide concentrations, we used the average of simulations in the lowest four layers of the model, between 5 to 100 meters. Given that $^{131}$I is highly radioactive, and $^{137}$Cs has a high solubility and a relatively long half-life, we will focus on their concentrations and deposition, respectively. Figures 5 and 6 illustrate the seasonal median of the 96-hour integrated $^{131}$I concentrations ($^{131}$I$^{intg\_conc\_seas}$, in units of ng m$^{-3}$) and of the maximum (total) $^{137}$Cs deposition ($^{137}$Cs$^{tot\_depos\_seas}$, in units of ng m$^{-2}$) within the study area.

Our mini-ensemble simulations show that the $^{131}$I$^{intg\_conc\_seas}$ values above $1.43\times10^{-3}$ ng m$^{-3}$ occur frequently in the cold period of the year, especially in fall, in the simulations of all members. This may be due in part to the lower (higher) PBLH in the

cold (warm) seasons and to the synoptic conditions. An exception is the distribution of $^{131}$I$^{intg\_conc\_seas}$ in the FNL-based simulations during the summer. This may be caused by a rare atmospheric circulation in summer, discussed later, and it may change as the modeling period is extended. In terms of spatial distribution, the $^{131}$I$^{intg\_conc\_seas}$ above $1.43 \times 10^{-3}$ ng m$^{-3}$ occur close to the source in the southeastern part of the domain, and the intensity of the $^{131}$I$^{intg\_conc\_seas}$ decreases with distance to the north. In other words, southeastern Qatar is the first area to be affected by dense $^{131}$I clouds in the event of a nuclear accident, especially during the cold period of the year. The advance of $^{131}$I$^{intg\_conc\_seas}$ above $1.43 \times 10^{-3}$ ng m$^{-3}$ in the south of Qatar in simulations based on FNL inputs occurs in both fall and winter, but is found only in winter in the GFS-based simulations. $^{131}$I$^{intg\_conc\_seas}$ values peak in the fall in both downscaled runs, with inputs from the ERA5- and FNL-WRF datasets, but the extent of high $^{131}$I$^{intg\_conc\_seas}$ values in the southeast of Qatar in fall is much larger in the ERA5-WRF run. This is also the case when comparing the FNL-WRF and FNL-based simulations. The differences have resulted in $^{131}$I$^{intg\_conc\_seas}$ values varying by a factor of 2 to 10 in the south of the area of interest between mini-ensemble members.

To investigate the influence of the particle release time on the radionuclide dispersion, the seasonal median of the particle release time (hours in local time (LT)) coinciding with the maximum concentration of $^{131}$I and the completion of $^{137}$Cs deposition is considered (contours in Figures 5 and 6). The results show that the highest $^{131}$I concentrations coincide with particles released between 9 a.m. and 2 p.m. LT in most parts of the study area. This is the time of day when the development of the planetary boundary layer, the intensification of the land-sea thermal gradient, and the resulting daytime onshore winds coincide in the region. Among the members, the earliest particle release times (between 6 and 9 a.m. LT) leading to the highest $^{131}$I concentrations are observed in simulations based on ERA5-WRF data.

All members have simulated $^{137}$C deposition above 10 ng m$^{-2}$ over a significant part of the study area during the cold period of the year. The highest $^{137}$Cs$^{tot\_depos\_seas}$ are observed in the south and southeast, near the emission point. Compared to the FNL- and GFS-based simulations, the simulations based on the FNL- and ERA5-WRF datasets show the much greater extent of the $^{137}$C deposition over almost the whole of Qatar in the winter and, to a lesser extent, in the fall. Among the mini-ensemble members, the highest $^{137}$Cs$^{tot\_depos\_seas}$ in the southeast are seen in the simulations based on ERA5-WRF inputs, followed by the FNL-WRF-based simulations, in fall. While the ERA5-WRF-based simulations of $^{137}$Cs$^{tot\_depos\_seas}$ exceed 30 ng m$^{-2}$, the deposition in GFS-based simulations is up to ten times lower in some points of the same area and same period. The $^{137}$Cs$^{tot\_depos\_seas}$ in the warm period of the year, except for the simulations based on the FNL dataset in summer, are mostly either close to or below 10 ng m$^{-2}$. The higher $^{137}$Cs$^{tot\_depos\_seas}$ in the cold period of the year, especially in winter, can be largely attributed to the seasonal increase of $^{137}$Cs transport by southerly winds, discussed later, and to the relative increase in wet deposition. The above results indicate that if a nuclear accident occurs during the cold period of the year, the magnitude of extreme $^{131}$I concentrations and $^{137}$Cs deposition in the south of Qatar may be up to 10 times greater than during the warm seasons. A similar order of magnitude in simulated variability is seen across the mini-ensemble members. With respect to the particle release times, $^{137}$Cs$^{tot\_depos\_seas}$ occur mainly when particles are released between 10 a.m. and 5 p.m. LT, in the presence of a turbulent boundary layer.

The analysis of the age spectra in subsection 3.2 shows the higher frequency of long-lived air parcels in the FNL-based simulations (see top row in Figure 2). To further examine these results, the spatial distribution of the full-year median of the 96-hour integrated $^{131}$I concentrations ($^{131}$I$^{intg\_conc\_full}$) and the full-year median of the air parcel ages coinciding with the maximum concentration of $^{131}$I found in each 96-hour run are examined (Fig. 7). As expected, the age of the Lagrangian particles decreases southward with proximity to the source. All mini-ensemble members simulate particles ages lower than 25

395 hours along with relatively high levels of $^{131}$I$^{intg\_conc\_full}$ at the southeastern corner of the study area. The longer-lived particles (above 35 hours) are found in higher latitudes. In simulations based on the FNL dataset, particle ages in northern Qatar exceed 40 to 50 hours, while they mostly do not exceed 40 hours in other mini-ensemble members. Furthermore, we obtain the larger extent of relatively higher $^{131}$I$^{intg\_conc\_full}$ to central Qatar in these simulations (considering both $^{131}$I$^{intg\_conc\_seas}$ and $^{131}$I$^{intg\_conc\_full}$ shown in Figures 5 and 7). Our analysis suggests that the longer-lived particles in FNL-based simulations can be attributed to

400 a higher number of air parcels reaching areas away from the source, rather than a lower transport speed.

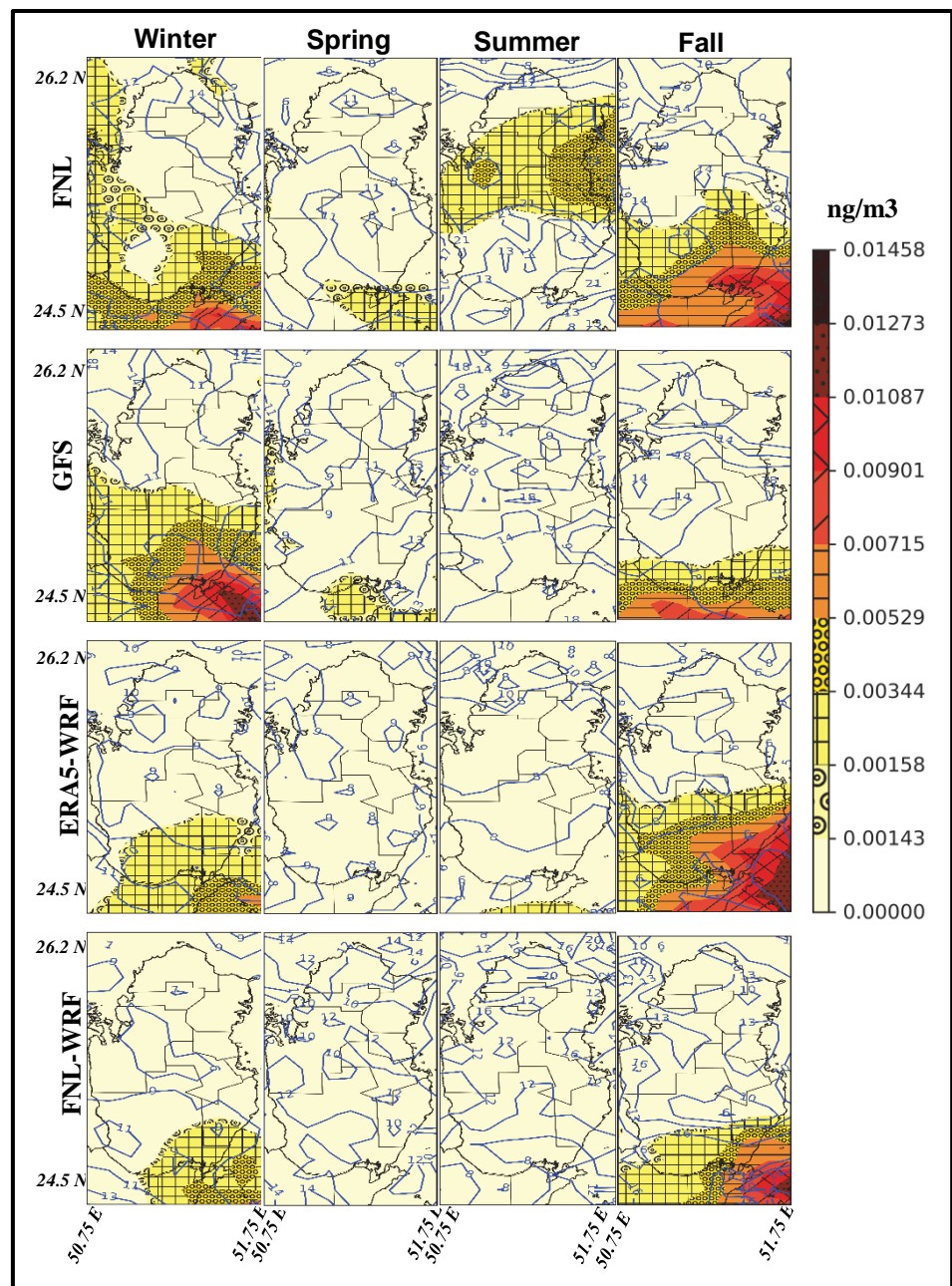

**Figure 5 Seasonal median of 96-hour integrated $^{131}I$ concentrations ($^{131}I^{intg\_conc\_seas}$). The contour lines (in local time, hours of the day) depict the seasonal median of the Lagrangian particle release time (local time) coinciding with the maximum $^{131}I$ concentrations found in each 96-hour run.**

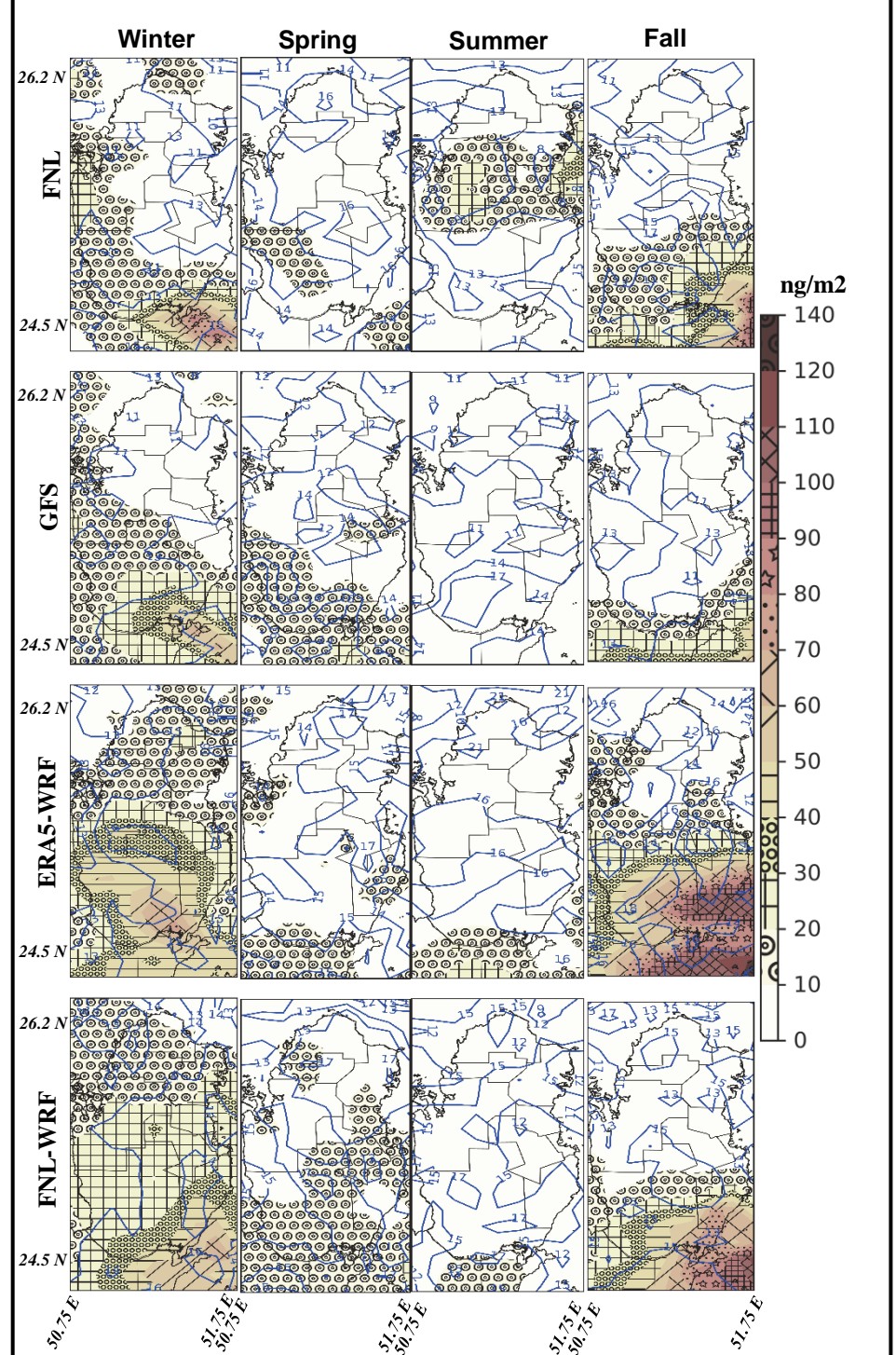

**Figure 6 Same as Figure 5, but for the seasonal median of total $^{137}$Cs deposition ($^{137}$Cs$^{tot\_depos\_seas}$) in the units of ng m$^{-2}$. The contour lines are the seasonal median of the Lagrangian particle release time (local time) coinciding with the completion of $^{137}$Cs deposition found in each 96-hour run.**

$^{131}$I$^{intg\_conc\_seas}$ and $^{137}$Cs$^{tot\_depos\_seas}$ are analyzed against the population density of Qatar (Fig. 8). The desert areas of southern and southeastern Qatar, over which radionuclides enter the country, host a small population or are almost uninhabited (Fig. 8-A). Figures 8-B and C show that the extremely high levels of $^{131}$I$^{intg\_conc\_seas}$ (greater than 0.6x10$^{-2}$ ng m$^{-3}$) and $^{137}$Cs$^{tot\_depos\_seas}$ (greater than 60 ng m$^{-2}$) occur mostly in areas with a population density of less than five persons per arc-second. In the populated areas (with a density of more than 8 persons per arc second) $^{131}$I$^{intg\_conc\_seas}$ and $^{137}$Cs$^{tot\_depos\_seas}$ do not exceed 0.4x10$^{-2}$ ng m$^{-3}$ and 40 ng m$^{-2}$ in most cases. Due to the exceptional weather pattern that occurs in the simulations based on the FNL dataset in the eastern part of Qatar (where the most densely populated areas are located) in summer, these simulations cause the highest values of $^{131}$I$^{intg\_conc\_seas}$ in densely populated areas. Otherwise, all mini-ensemble members simulate the highest $^{131}$I$^{intg\_conc\_seas}$ during the cold seasons. The highest $^{137}$Cs$^{tot\_depos\_seas}$ in the same areas are found based on ERA5 and FNL-WRF in the cold period of the year.

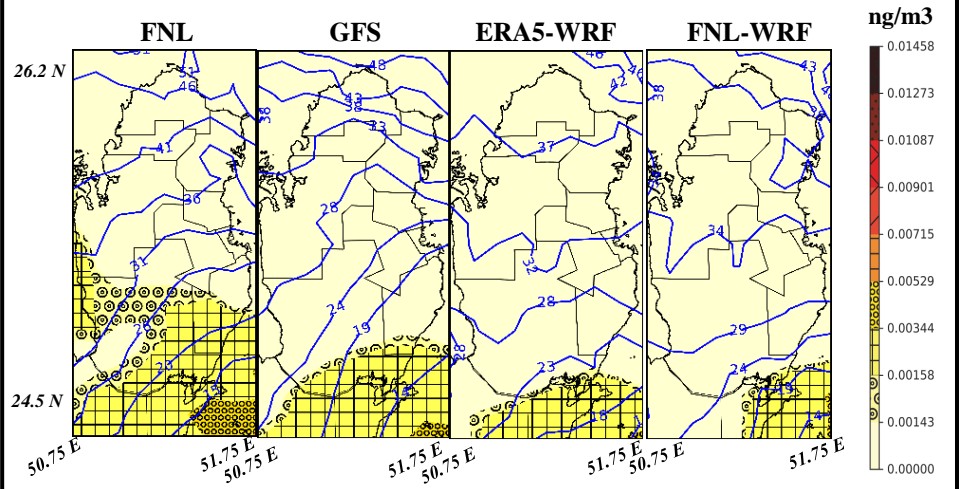

**Figure 7 The full-year median of 96-hour integrated $^{131}$I concentrations ($^{131}$I$^{intg\_conc\_full}$). The contour lines are the full-year median of age spectra coinciding with the maximum $^{131}$I concentrations found in each 96-hour run.**

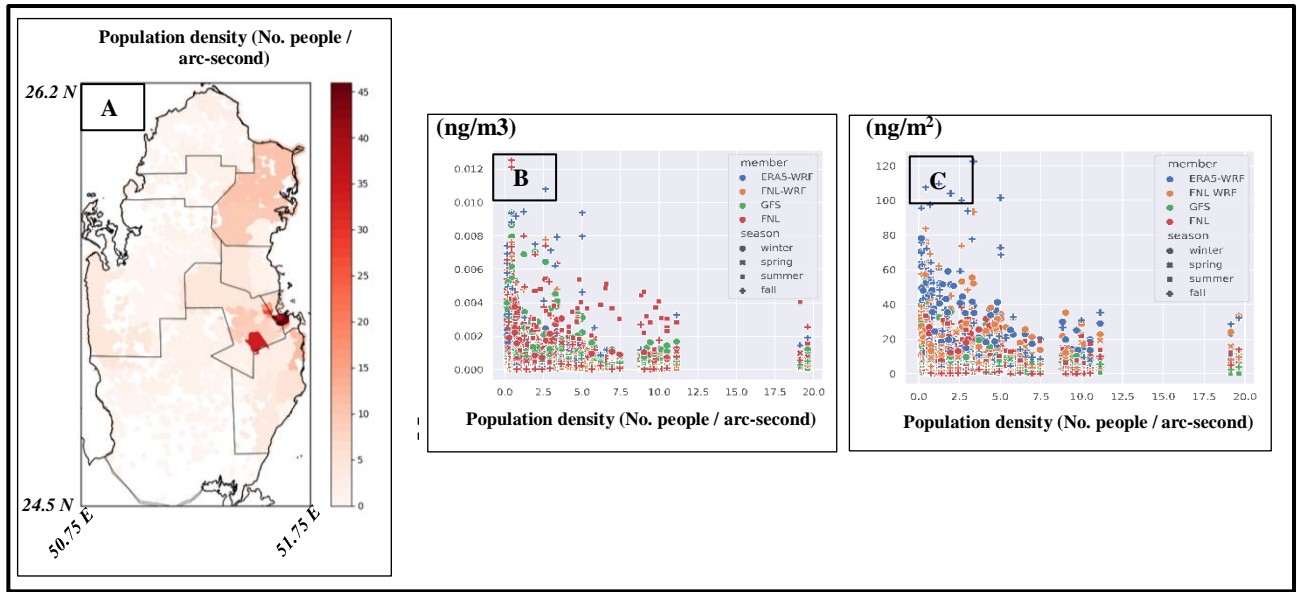

**Figure 8 A: The gridded population density of Qatar at one arc-second resolution in 2020. The relationship between** $^{131}I^{intg\_conc\_seas}$
**(ng m$^{-3}$) and** $^{137}Cs^{tot\_depos\_seas}$ **(ng m$^{-2}$) and the population density of Qatar are shown in B and C, respectively. The shapes and colors**
**of the markers represent seasons and mini-ensemble members.**

In addition to the magnitude of the transported radioactive materials, the temporal distribution of the extreme events in the
affected area is also of importance for preparedness programs. Figure 9-A shows the frequency of occurrence (FoO) of $^{131}I$
concentrations above the 66th percentile. In all members of the mini-ensemble, more than half of the events, especially in the
northern part, take place in the winter. While the FoO of high $^{131}I$ varies between around 15 and 30% in spring, the lowest FoO
is observed in the summer when less than about 10% of the extreme events occur. The seasonal FoO of $^{137}Cs$ concentrations
also show very similar results (S14). A case where ERA5-WRF simulated the northwestward movement of the near-surface
$^{131}I$ concentrations (ng m$^{-3}$) in January at noon is shown in the upper panel of Figure 9-B. We find a similar pattern in a large
number of events in which high levels of radionuclides are transported to Qatar. This synoptic pattern is related to the
juxtaposition of low and high-pressure cells located to the west and east of the region. The resulting pressure gradient cause
strong south/southeast winds to develop between two cyclonic and anti-cyclonic cells, bringing the dense $^{131}I$ clouds into the
study area. This pattern mainly occurs in the late winter and early spring, coinciding with the southward movement of the
westerlies and the eastward movement of the Saudi Arabian subtropical high pressure system (De Vries et al., 2016). The
lower panel of Fig. 9-B shows the summertime mean of near-surface $^{131}I$ concentrations (ng m$^{-3}$), obtained from simulations
based on the ERA5-WRF database. The near-surface atmospheric circulation is superimposed on these simulations. The
seasonal pattern found here illustrates well why very few extreme events are observed in the study area during the summer.
The northwest-southeast winds, known as the Shamal winds (Yu et al., 2016), cause the simulated radionuclides to move away

from the study area. Therefore, the highest (lowest) contamination risk to the population of Qatar from the occurrence of a nuclear accident in B-NPP and the resulting release of radionuclides is expected to occur in winter (summer).

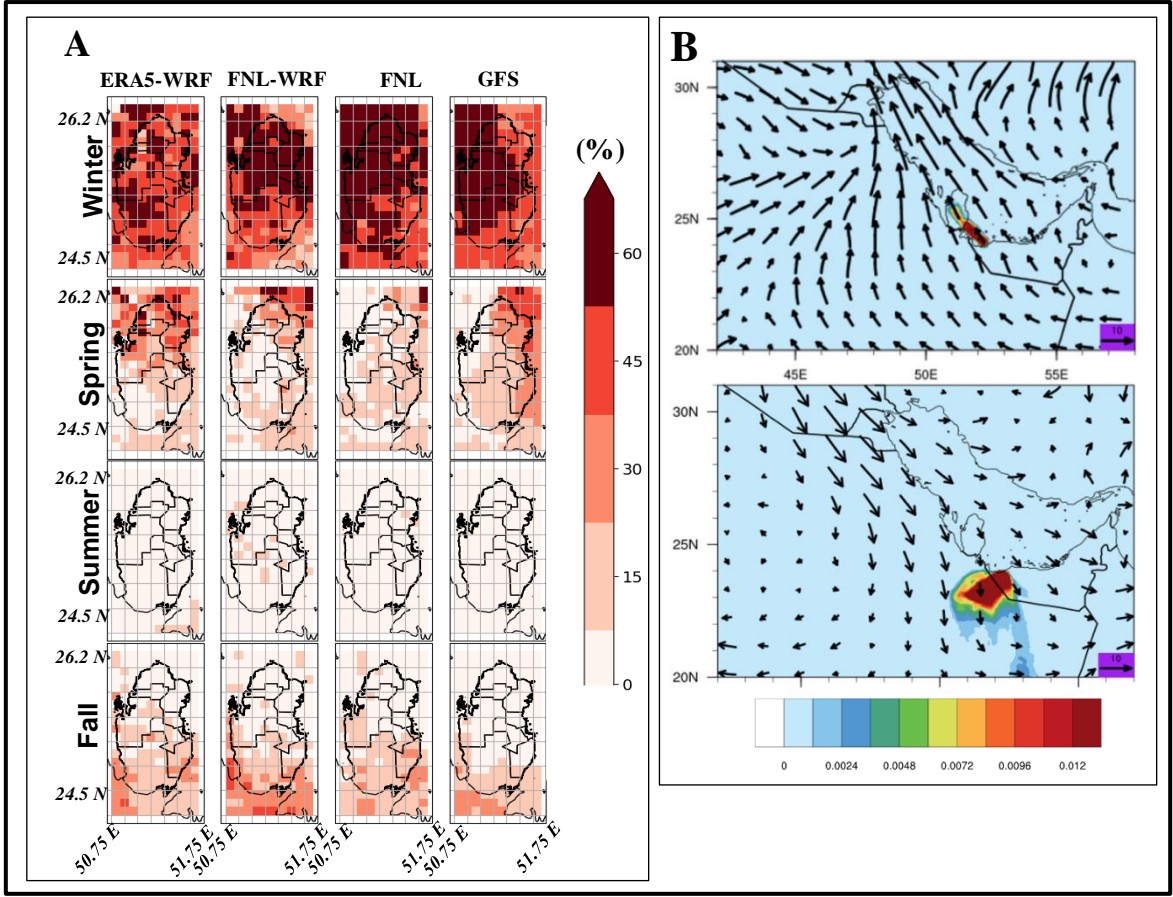

Figure 9 A: Frequency of occurrence (%) of $^{131}$I concentrations above the respective 66th percentile. B: The figure at the top shows the simulation of near-surface $^{131}$I concentrations (ng m$^{-3}$) based on the ERA5-WRF dataset on January 14, 2019. The figure at the bottom is the summer average of near-surface $^{131}$I concentrations. The wind field (m/s) at 12:00 on the first day of the 96-hour simulation period is used to represent the atmospheric circulation during the radionuclide transport.

**3.4 Sensitivity of simulations to turbulence scheme, emission profiles, and emission location**

This section focuses on investigating the sensitivity of radionuclide simulations using ERA5-WRF inputs through four distinct sensitivity runs. Each of these runs explores specific variations in model setup, such as turbulence scheme, emission duration, emission height, and emission location. To establish a baseline, all sensitivity runs are compared against the control run that utilizes GTM as the turbulence scheme and emission characteristics that are explained in the subsection in 2.3. Results show that the use of the skewed turbulence model (STM) leads to a more frequent occurrence of high $^{131}$I concentrations and $^{137}$Cs deposition within Qatar (Fig. 10). The upper quartiles of the STM-based $^{131}$I concentrations are around 10% and 16% higher

than those of the GTM-based, control simulations in winter and year-round, respectively. The upper quartiles of STM-based deposition simulations also increase by 23% and 12% over the same periods in comparison to GTM-based, control deposition simulations. As noted previously, the elevation in $^{131}$I concentrations and $^{137}$Cs deposition based on STM is not surprising, given that STM promotes more downdrafts than updrafts. This pattern leads to higher surface concentrations and deposition
in the proximity of pollution sources Pisso et al. (2019).

In the CRST run with the extension of the release duration from 24 hours to 96 hours, a decrease in the amount of $^{131}$I concentrations and $^{137}$Cs deposition compared to the control run is observed, except for $^{131}$I concentrations in spring. This result is to be expected, as the particles released at the end of the 96-hour simulation period have less time to reach the study area, resulting in a considerable decrease in the concentration/deposition of radioactive materials. For instance, this
implementation leads to a reduction of approximately 75% and 45% (170% and 140%) of the upper quartiles of $^{131}$I concentrations ($^{137}$Cs deposition) in winter and fall, compared to the control run.

Morino et al. (2011) used the Eulerian chemical transport model, CMAQ, to investigate the sensitivity of $^{131}$I and $^{137}$Cs deposition to the vertical profile of emissions from the Fukushima nuclear accident. They concluded that the fractions of $^{131}$I and $^{137}$Cs deposition were insensitive to the change in emission height. In contrast, our simulations indicate that an increase in
emission height leads to a significant decrease in $^{131}$I concentrations and $^{137}$Cs deposition, except in the summer. Specifically, the upper quartile of $^{131}$I concentrations in the RHST run decreases by approximately 25% and 130% in the winter and fall, respectively, compared to the control run. Similarly, the increase in emission height is associated with a decrease in $^{137}$Cs deposition, with the upper quartile of simulations showing a considerable decrease of up to 65% in RHST simulations compared to the control implementation during the fall. The severity of deposition reduction lessens during the winter and
spring seasons, and is instead accompanied by a slight increase during the summer. It seems that increasing the release height results in particle entry into the upper layers of the atmosphere, especially during the cold months when regional and larger-scale atmospheric patterns are more prevalent. This results in the transportation of the majority of radionuclides to more distant locations, leading to a decrease in $^{131}$I concentrations and $^{137}$Cs deposition.

The RLST run provides insights into the impact of shifting the emission point from B-NPP to Bu-NPP. The results demonstrate
notable disparities in the transportation of radioactive materials to Qatar. Specifically, the result reveal that in the event of a nuclear accident at Bu-NPP, there is a relatively reduced transfer of radioactive materials to Qatar compared to emissions originating from B-NPP. This trend holds true for most of the year. This outcome can indeed be attributed to the greater distance between Bushehr NPP and Qatar, as anticipated. The upper quartile of the full-year median of $^{131}$I concentrations and $^{137}$Cs deposition in RLST simulations decrease by a factor of approximately 2.2 and more than 3, respectively. The seasonal
variability of the transport and deposition of radioactive materials from Bu-NPP is a noteworthy aspect to consider in our analysis. the simulations of the RLST run show that $^{137}$Cs deposition is more prominent in winter and spring compared to other seasons. Moreover, the upper quartile value of $^{131}$I concentrations from Bu-NPP exceeds that of B-NPP by more than 80% in spring which is in contrast to the simulations of other seasons.

In Figure 11-A, a distinct spatial pattern emerges in the Bu-NPP simulations when compared to the B-NPP simulations (Figs. 5, 6, and 7). The simulated $^{131}I^{intg\_conc\_full}$ exhibit relatively high values in the northern and eastern regions, including Doha, particularly during the winter and spring seasons. This occurrence can be attributed to the downward movement of westerlies, facilitating the transport of air masses enriched with $^{131}I$ towards these areas. The full-year median of the total $^{137}Cs$ deposition ($^{137}Cs^{tot\_depos\_full}$) (Fig. 11-B) closely follows this temporal pattern, showing prominent peaks during the spring and winter seasons within the northern and eastern regions of Qatar. Conversely, during the fall and summer seasons, the levels of $^{131}I^{intg\_conc\_full}$ and $^{137}Cs^{tot\_depos\_full}$ are greatly reduced. The age of particles corresponding to high levels of $^{131}I$ and $^{137}Cs$ simulations indicates their earlier entry into the northern half of Qatar during these two seasons, reaching the region within approximately 40-50 hours and 60 hours after release, respectively.

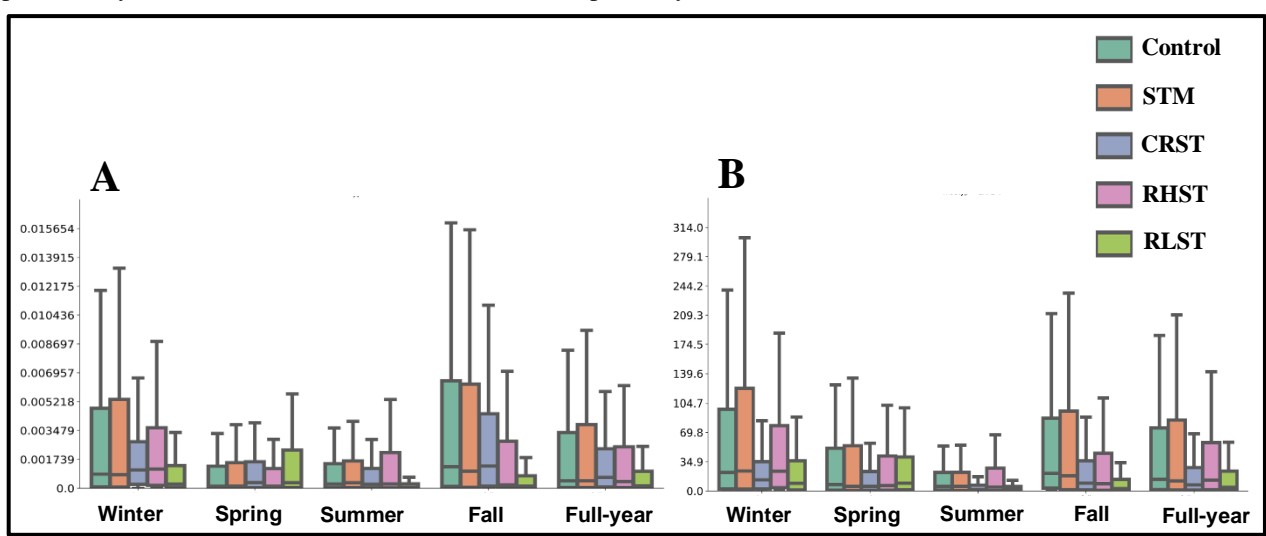

**Figure 10 A: These boxplots depict the 96-hour integrated simulations of $^{131}I$ concentrations (ng m$^{-3}$) based on ERA5-WRF inputs. The simulations include control runs shown in cyan, and sensitivity runs with the skewed turbulence model (STM) in brown, increased release duration from 24 to 96 hours (CRST) in blue, modified release height from the 100-300m agl to the 100-700m agl (RHST) in pink, and a change in the release location from Barakah to Bushehr nuclear power plant (RLST) in green. B: The boxplots show the same for the total deposition of $^{137}Cs$ (ng m$^{-2}$). The quartiles of the simulations are depicted by the box borders, while the whiskers illustrate the range between 1.5 times the interquartile range above the upper quartile and below the lower quartile.**

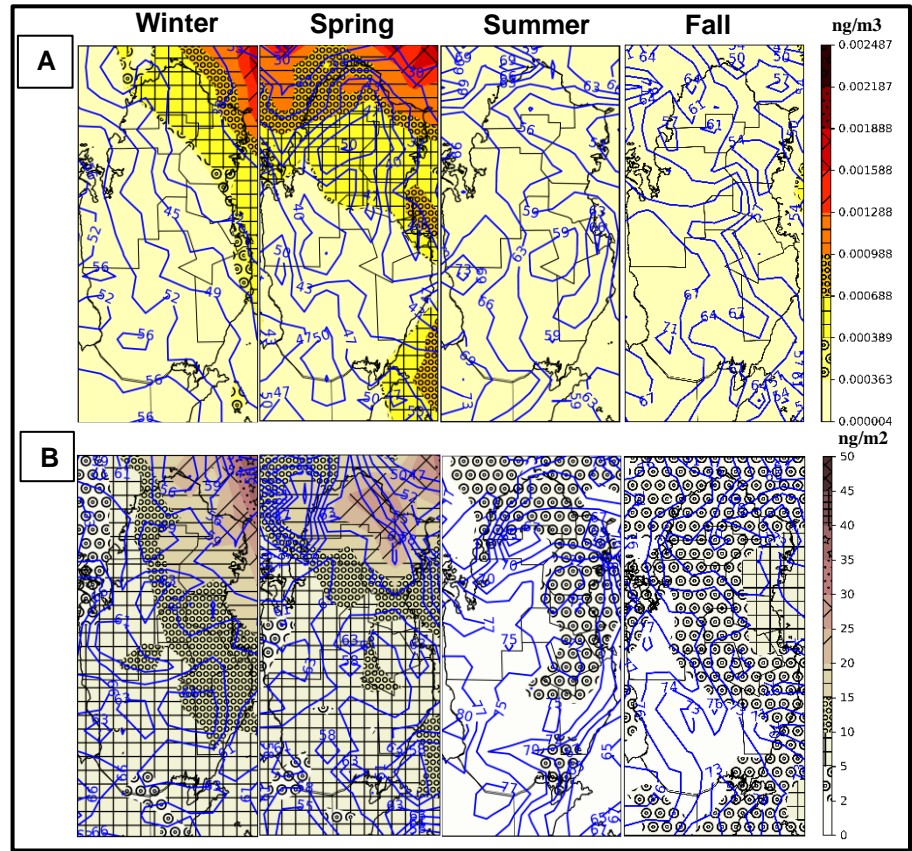

**Figure 11: Simulated $^{131}$I$^{intg\_conc\_full}$ (A) and $^{137}$Cs$^{tot\_depos\_full}$ (B) based on ERA5-WRF inputs originating from Bushehr NPP. The contour lines are the full-year median of age spectra coinciding with the maximum $^{131}$I concentrations and the completion of $^{137}$Cs deposition found in each 96-hour run.**

## 3.5 Inter-comparison of mini-ensemble members

Radionuclide simulations based on (re)analysis datasets are assumed to be a better approximation of the actual atmospheric conditions than those based on the forecasting dataset (GFS, in this study) (Leadbetter et al., 2022). The statistical evaluation metrics used here are recommended by Maurer et al. (2018), who assessed transport models simulating Xe-133 using measurements from six International Monitoring System (IMS) stations. In addition to other common evaluation metrics, they used the fractional bias (FB) and the fraction within a factor of 5 (F5). FB, in the range -2 and 2, is the bias of the simulated mean values normalized by the sum of the simulation and measurement means and multiplied by 2. F5 is the fraction of simulations that are at most one factor larger (5) or smaller (0.2) than the reference values. The Spearman correlation coefficient (r) calculated between the simulations of $^{131}$I concentrations shows that the GFS-based simulations are closely associated with those of the (re)analysis-based members (red circles in the bottom row in Figure 12). The GFS-based simulations attain the highest correlation with the FNL-based simulations (0.85) followed by FNL-WRF-based simulations (0.73). While the GFS-

based simulations have (on average) a small FB (0.07) compared to the FNL-based simulations, they are positively biased compared to the simulations based on the FNL-WRF (-0.99) and ERA5-WRF (-0.98) datasets. According to F5, 74.06% and 55.69% of the GFS-based simulations are within a factor of 5 of the FNL and FNL-WRF-based simulations, respectively, whereas F5 decreases to 54.72% between the GFS- and ERA5-WRF-based simulations. The RMSE between simulations based on the GFS and FNL datasets ($1.49 \times 10^{-2}$ ng m$^{-3}$) is smaller than that found between the former and the FNL-WRF- (0.0235 ng m$^{-3}$) and ERA5-WRF-based ($2.41 \times 10^{-2}$ ng m$^{-3}$) simulations. As for the other metrics, the GFS-based simulations produce the lowest and highest NMSE against the FNL (1.24) and ERA5-WRF (8.78) simulations, respectively. In short, the GFS- and FNL-based simulations have the highest agreement due to the large similarities between their meteorological inputs (subsection 2.2). Among all mini-ensemble members, the largest difference occurs between simulations based on ERA5-WRF and FNL datasets (r=0.55, FB=-0.92, F5=50.6%, RMSE=$2.62 \times 10^{-2}$ ng m$^{-3}$, and NMSE=11.18). On the other hand, the downscaling of the inputs and the application of the same simulation code yield higher agreement of the simulations based on the FNL-WRF and ERA5-WRF datasets than to those based on the FNL and FNL-WRF datasets (r=0.7 vs. 0.67, FB=0.01 vs. -0.93, F5=69.83% vs. 53.3%, RMSE=$0.78 \times 10^{-2}$ ng m$^{-3}$ vs. $2.48 \times 10^{-2}$ ng m$^{-3}$, and NMSE=2.74 vs. 10.08). These results point out that particle dispersion modeling is primarily influenced by both the meteorological inputs and the dispersion model of choice, which is consistent with the results of Karion et al. (2019).

The evaluation metrics of $^{137}$Cs concentrations (blue squares in the top row in Figure 12) lead to the same results as discussed above. For example, the simulations based on the GFS dataset show the highest agreement with the FNL-based simulations (r=0.85, FB=0.07, F5=74.98%, RMSE= 2.64 ng m$^{-3}$, and NMSE=1.19). Consistent with our findings, the simulations derived from FNL and ERA5-WRF datasets exhibit the most robust agreement, while the simulations based on FNL-WRF and ERA5-WRF datasets demonstrate the least agreement among the ensemble members relying on the (re)analysis inputs. The GFS-based simulations mostly lead to closer agreement with those based on the (re)analysis datasets when comparing the $^{137}$Cs concentration simulations than the $^{131}$I concentration simulations.

The assessment of the simulated $^{137}$Cs deposition reveals a notable lack of agreement between all mini-ensemble members, as depicted in Fig. 13. In particular, the disparities in wet deposition rates are considerably more pronounced than those in dry deposition rates. This can be largely explained by the high degree of uncertainty associated with the inputs required to calculate wet deposition, such as precipitation occurrence and rate and cloud water content. These uncertainties contribute to the overall uncertainty in the deposition parameterization (Gudiksen et al., 1988), thereby leading to inconsistencies in the wet deposition rates, which are more pronounced than the dry deposition rates. Despite the crucial role of precipitation in modeling, our mini-ensemble members exhibited poor performance in simulating precipitation values throughout the region and specifically in Qatar (subsection 3.1). Moreover, FLEXPART and FLEXPART-WRF use different scavenging schemes (see subsection 2.1). However, similar to the simulated concentrations, GFS-based simulations of $^{137}$Cs (dry/wet) deposition show closest agreement with FNL-based simulations. We also find that the largest difference between the simulated dry deposition of $^{137}$Cs from all mini-ensemble members occurs between those forced by the FNL and ERA5-WRF datasets. Better agreement between the

simulations based on the FNL-WRF and ERA5-WRF datasets than between those based on the FNL and FNL-WRF datasets

is also obtained for the simulations of $^{137}$Cs dry deposition, but not for the simulations of $^{137}$Cs wet deposition. The above results also apply to the simulations of $^{131}$I deposition (Fig. S15), except that in this study $^{131}$I is assumed to be insoluble, remaining in the gas phase and not subject to wet deposition.

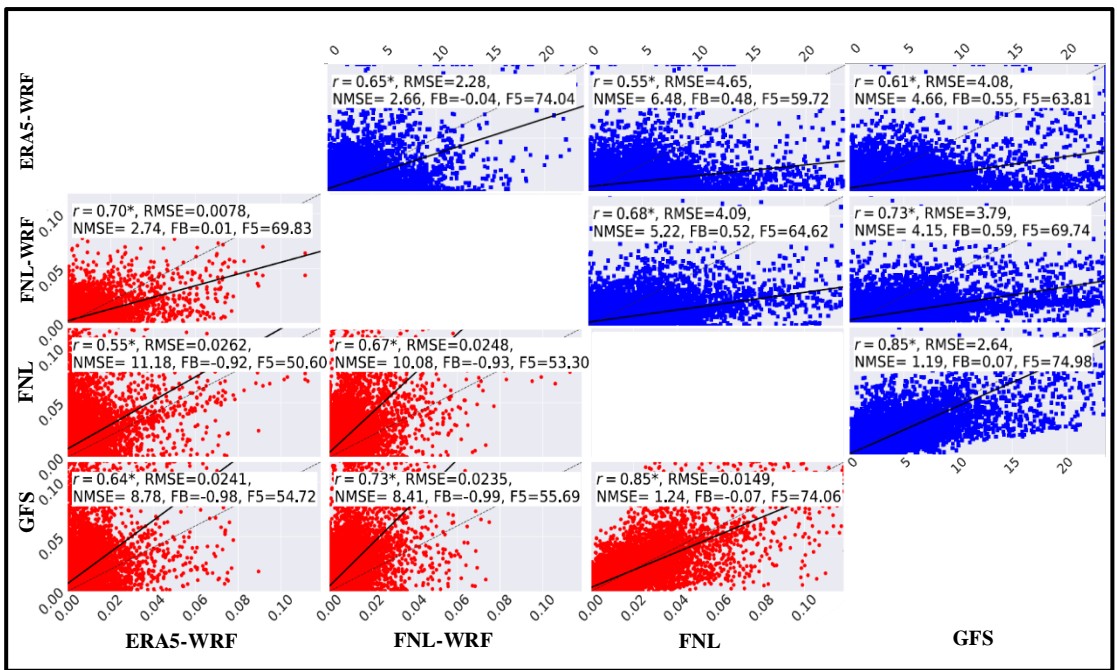

**Figure 12 The inter-comparison of the 96-hour integrated simulations of near-surface $^{131}$I (red circles) and $^{137}$Cs (blue squares)**
**concentrations (ng m$^{-3}$) simulated at each grid point at each day of 2019. Solid and dotted lines show regression and identity lines, respectively. The asterisk next to the Spearman correlation coefficient (r) indicates a statistically significant correlation at p<0.05.**

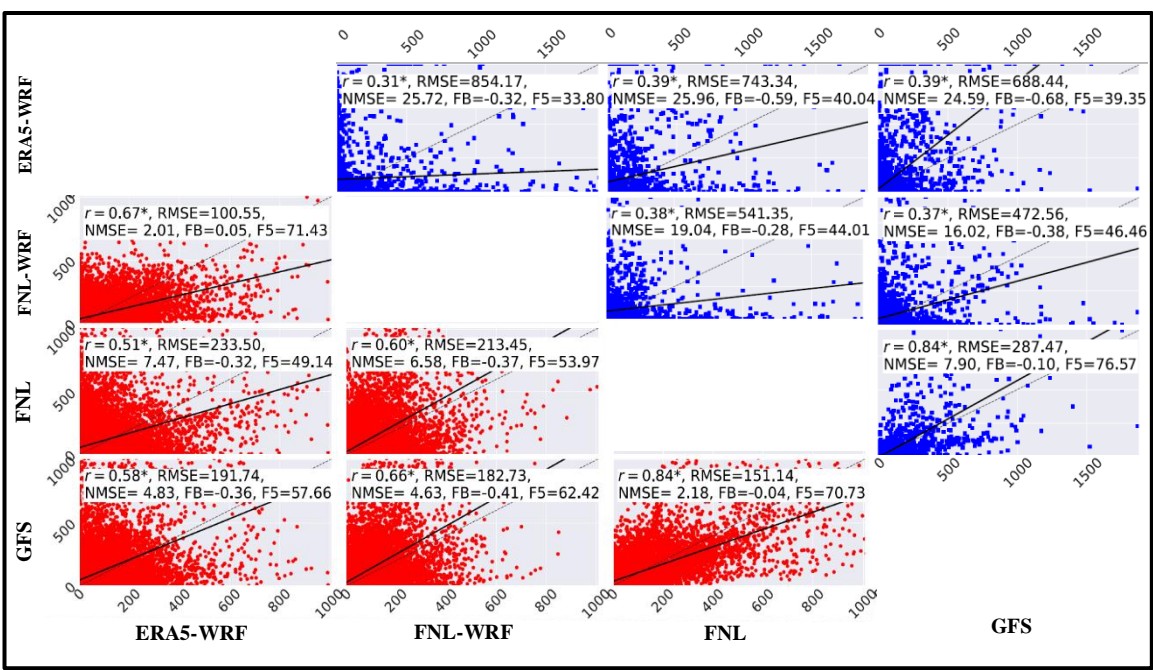

**Figure 13 Same as Figure 12 but for $^{137}$Cs dry (in red circles) and wet (in blue squares) deposition (ng m$^{-2}$).**

## Conclusions

In this study, we examined the spatio-temporal dispersion of radionuclides, including $^{131}$I and $^{137}$Cs, in the fictitious event of nuclear accidents at the Barakah Nuclear Power Plant (B-NPP) in the United Arab Emirates (UAE). The resulting concentrations and deposition of the studied radionuclides were simulated using the Lagrangian particle dispersion model FLEXible PARTicle (FLEXPART) and FLEXPART coupled with the Weather Research and Forecasting model (FLEXPART-WRF). To investigate the diurnal and seasonal variations of the radionuclide dispersion, the particles (air parcels)

were released within the first 24 hours of a 96-hour simulation period at B-NPP, between 100 and 300 m above ground level, iterated daily over the year 2019. The source term is scaled to the maximum estimates of the radioactivity emissions from the Fukushima accident (6.9 kg (22 PBq) of $^{137}$Cs and 0.042 kg (192 PBq)) for the purpose of providing a realistic benchmark for the reference values. We found differences in the simulations with respect to the meteorological inputs. We investigated the meteorological uncertainties by constructing a mini-ensemble with three members based on (re)analysis datasets including

Final Analysis (FNL) at native resolution, FNL and the ECMWF 5th Generation Reanalysis (ERA5) downscaled by WRF (ERA5- and FNL-WRF), and one member based on the Global Forecast System (GFS) run by NCEP. We compared the daily means of surface wind speed and temperature and daily total precipitation from the above datasets with observations measured from 157 monitoring stations within the model domain. This mini-ensemble provided the basis for comparing the simulations based on the forecast and (re)analysis datasets. The FNL- and GFS-simulations were compared with the simulations based on

the FNL-WRF and ERA5-WRF datasets to determine the impact of downscaling and of using different/same model simulation codes (FLEXPART vs. FLEXPART-WRF) on the modeled dispersion. We conducted a sensitivity analysis on ERA5-WRF-based simulations, exploring the effects of variations in turbulence scheme under convective conditions, emission duration, emission height, and emission location. We examined the simulations of all four members in relation to Qatar's population density, with the goal of identifying possible risks to populated areas from a nuclear accident in the region. A summary of the

results of the study is presented below:

1- **Transport of radionuclides:** To analyze the time interval between emission and exposure to radionuclides, we investigated the age composition of the radionuclide plumes. Analysis of air parcel ages indicates that dense radionuclide clouds arrive in the south of the study area approximately 20 to 30 hours after the emission. A significant part of $^{131}$I released is transported to the most distant parts of the study area up to 40 to 50 hours after the accidents.

All members simulated that a large fraction of the $^{137}$Cs deposition occurs within the first 75 to 80 hours after the emission. The largest deposition of longer-lived particles was found in FNL-based simulations for all seasons, except in the fall. We attribute this to the more distant transport of air parcels from the emission point in FNL-based simulations, compared to other members. The two members which are forced by the downscaled datasets (FNL-WRF and ERA5-WRF) simulated a more similar distribution of air parcel ages than those simulated by the other two

members.

2- **Distribution of extremely high concentrations and deposition of radionuclides:** We calculated the seasonal median of 96-hour integrated $^{131}$I concentrations ($^{131}$I$^{intg\_conc\_seas}$, in units of ng m$^{-3}$) and the total $^{137}$Cs deposition ($^{137}$Cs$^{tot\_depos\_seas}$) across the study area. As expected, all members simulated much higher $^{131}$I$^{intg\_conc\_seas}$ over the south/southeast of Qatar, which is the closest point of the study area to the emission point. The inter-seasonal

comparison of the simulations showed that the mini-ensemble members simulated the largest advance of $^{131}$I$^{intg\_conc\_seas}$ from the source to south/southeast of Qatar in the cold period of the year. The simulations of $^{131}$I$^{intg\_conc\_seas}$ between the members of the mini-ensemble differed by a factor of 10. The differences in the $^{131}$I$^{intg\_conc\_seas}$ simulations based on FNL and FNL-WRF datasets demonstrated how the use of different model simulation codes and the downscaling of meteorological inputs can affect the FLEXPART modeling and, consequently, the decisions made based on its

simulations after nuclear accidents. Similarly, remarkable differences were found in the spatio-temporal distribution of $^{131}$I$^{intg\_conc\_seas}$ simulations based on the FNL and GFS datasets. This is the case even though these datasets are produced by the same base meteorological model. We attribute this to the fact that differences, however small, in the meteorological inputs that lead to cumulative deviations in the transport and concentration calculations of atmospheric pollutants. As with $^{131}$I$^{intg\_conc\_seas}$, $^{137}$Cs$^{tot\_depos\_seas}$ simulations from all the mini-ensemble members peaked in the

southeastern part of the study area in the cold period of the year when both wet and dry deposition occur. The largest expansion of $^{137}$Cs$^{tot\_depos\_seas}$ above 10 ng m$^{-2}$ were found in the simulations based on the ERA5-WRF and FNL-WRF datasets in winter. The highest levels of $^{137}$Cs$^{tot\_depos\_seas}$ (above 30 ng m$^{-2}$) were found in the simulations based on the

ERA5-WRF dataset in the southeastern corner of Qatar in the fall. This region received far less $^{137}Cs^{tot\_depos\_seas}$ (around 10 ng m$^{-2}$ and less) in the simulations based on the GFS and FNL datasets in the same period. The examination of the release time of the air parcels resulting in the extreme $^{131}I$ concentrations and $^{137}Cs^{tot\_depos\_seas}$ showed that the corresponding particles are mostly released between 9 a.m. and 2 p.m. LT. The development of the boundary layer height, the intensification of the thermal gradient between the land and sea, and the resulting onshore winds increase the transport of radionuclides to the study area during this time of day. The analysis of the frequency with which $^{131}I$ and $^{137}Cs$ concentrations above the 66$^{th}$ percentile are transported to the populated areas of eastern Qatar showed that, for all members, more than 50% of the extreme cases occur in winter and between 15% and 30% in spring. The above results indicate that any nuclear accident in the winter will more likely be accompanied with the highest radionuclide concentrations and deposition within the study area. This pronounced intra-annual variation is attributed to a seasonal atmospheric pattern in which south/southeast winds transport the dense radionuclide clouds. The collocation of population density showed that the populated areas (with more than 8 persons per arc-second) receive moderate to low levels of $^{131}I^{intg\_conc\_seas}$ and $^{137}Cs^{tot\_depos\_seas}$ (below 0.4x10$^{-2}$ ng m$^{-3}$ and 40 ng m$^{-2}$). Uninhabited areas in southern Qatar receive the highest levels of $^{131}I^{intg\_conc\_seas}$ and $^{137}Cs^{tot\_depos\_seas}$ (above 0.6x10$^{-2}$ ng m$^{-3}$ and 60 ng m$^{-2}$).

3- **Sensitivity of simulations to model parameters and emission characteristics:**

We analyzed the sensitivity of radionuclide simulations based on ERA5-WRF dataset to four variations of model setup. Results show that the use of the skewed turbulence model (STM) leads to a more frequent occurrence of high $^{131}I$ concentrations and $^{137}Cs$ deposition within Qatar. Increasing emission height leads to a significant decrease in $^{131}I$ concentrations and $^{137}Cs$ deposition, particularly during the cold period. Extending the emission duration causes a decrease in the amount of $^{131}I$ concentrations and $^{137}Cs$ deposition. As anticipated, due to the longer distance, the change in the emission point from B-NPP to Bushehr-NPP (Bu-NPP) results in reduced transfer of radioactive materials to Qatar, except during the spring season. The simulations from Bu-NPP exhibit distinct spatial patterns compared to those from B-NPP. Concentrations of $^{131}I$ peak in the northern and eastern regions during winter and spring, which can be attributed to a southward shift of the westerlies. The deposition of $^{137}Cs$ follows a similar pattern. The entry of particles inducing high intensities of $^{131}I$ and $^{137}Cs$ over the northern region of Qatar occurs within approximately 40-50 hours and 60 after their release, respectively. The significant changes observed in the simulations during the sensitivity analysis can be attributed to the promotion of more downdrafts than updrafts with the STM, particle entry into the upper layers of the atmosphere with increasing emission height, less time for particles to reach the study area with extended emission duration, and the greater distance between the emission source and catchment area and along with the varying impact of atmospheric patterns on the transport of radionuclides with a change in emission point.

4- **Inter-comparison of mini-ensemble members:** The simulations of $^{137}Cs$ and $^{131}I$ concentrations based on the GFS dataset were found to have the closest agreement with the simulations based on the FNL dataset because they share a

common meteorological base model. The comparison of wind speed, precipitation, and temperature from the GFS and FNL datasets showed the best agreement compared to the other two input datasets. However, we also found important differences in the spatio-temporal distribution of GFS- and FNL-based simulations. This may be due to the cumulative effect of differences in the meteorological inputs on particle dispersion. The comparison of simulations

based on ERA5- and FNL-WRF indicated that the downscaling of the inputs and the application of the same simulation increases the agreement of resulting simulations to an extent that exceeds the degree of similarity between simulations based on FNL and FNL-WRF, with the same source of meteorological inputs. The deposition simulations of all members showed relatively large inconsistency for both radionuclides. This was more pronounced for the simulations of $^{137}$Cs wet deposition. This is in part because wet deposition forcing factors are among the most

challenging meteorological parameters to model accurately. Moreover, the recently updated wet deposition scheme implemented in the FLEXPART uses different methods to determine the occurrence of wet deposition than FLEXPART-WRF (Girard et al., 2016, Gudiksen et al., 1988, Evangeliou et al., 2017).

**Data availability.** The FLEXPART and FLEXPART-WRF simulations are available upon request. The open-source codes for

the FLEXPART 10.4 and FLEXPART-WRF 3.3.2 can be downloaded from https://www.flexpart.eu/downloads (last access: 27 May 2022). Qatar's high-resolution population density datasets are freely available at https://data.humdata.org/dataset/qatar-high-resolution-population-density-maps-demographic-estimates (last access: 27 May 2022).

**Author contributions.** SON performed the WRF, FLEXPART, and FLEXPART-WRF simulations and led the integration of

results and writing. SON and TC designed the experiments. All co-authors have read the paper and provided professional comments.

**Competing interests.** The authors declare that they have no conflict of interest.

**Acknowledgments.**

The CyI High-Performance Computing Facility and Qatar Environment and Energy Research Institute (QEERI) High-

Performance Computer provided computational resources supporting this work. The authors thank the FLEXPART and FLEXPART-WRF developers for providing the transport model source codes. We also acknowledge the efforts of the editor and anonymous referees, who thoroughly reviewed the manuscript and provided valuable feedback that helped us to strengthen the arguments and results presented in the paper. Their contribution was critical in ensuring the soundness and rigor of our work.

**Financial support.**

This research has received funding from the Qatar Environment & Energy Research Institute (QEERI), an entity of the Hamad Bin Khalifa University, wholly owned by the Qatar Foundation for Science, Education, and Community Development.

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
