# Peer review of "Spatio-temporal variation of radionuclide dispersion from nuclear power plant accidents using FLEXPART mini-ensemble modeling"

_Atmospheric Chemistry and Physics, 2022_

## Author Response (AR1)

Dear Dr. Brioude, referee #1, and referee #2,

We would like to thank you for the time and effort dedicated to providing feedback on our manuscript. We believed that the revised submission has undergone a considerable improvement in terms of explaining the study necessity and achievements thanks to the constructive comments and suggestions made by referees #1 and #2. All page numbers in our responses refer to the revised manuscript file without tracked changes. In the following, we list our responses to all comments and highlight the changes implemented in the revised manuscript.

**Our responses to all comments made by referee #1 are as follows:**

- I do not see any novelty neither in methods nor in results or discussion. I do not mean to degrade authors' great work, but I do not see this manuscript of relevance for publication in ACP.

  - ✓ **The novelty of this paper lies in the application of the FLEXPART Langrangian model for radionuclide dispersion simulations to inter-compare the variability and quantify uncertainties using a broad ensemble of (re-)analyses, as well as forecasted meteorological input datasets.**

  - ✓ **There are several works previously published in ACP using FLEXPART with a single meteorological dataset (Zhu et al., ACP 2020)(Sauvage et al., ACP 2017). In this study, we have designed a four-member ensemble that allows for the first time to inter-compare simulations produced by FLEXPART and FLEXPART coupled with the Weather Research and Forecasting model (FLEXPART-WRF) and capture the effect of downscaling on FLEXPART dispersion modelling. We evaluate the relative performance of forecast runs against three re-analysis runs (Table 1), both complementing and extending the approach (taking a reanalysis run as a reference run) by Leadbetter et al. (ACP, 2022).**

  - ✓ **Our paper also expands upon previous studies published in ACP that focus on a fictitious case to quantify the impact risks. For instance, the study by Salminen-Paatero et al. (ACP, 2020) uses dispersion modeling results with the SILAM model to study due to hypothetical reactor accidents in Finland. Our methodology of**

**continuous release allows us to uniquely estimate the probability of occurrence over each hour-of-day and month-of-the-year.**

✓ **In summary, our paper is not devoted to the model performance analysis in a specific real-world accident, but rather we aim to highlight and quantify the strong variability due to diurnal and seasonal meteorological variations stemming from the choice of re-analysis used for dispersion modelling. We capture the range of uncertainty through the iterative multi-day simulations, starting each day of the year, and the analysis of the resulting age spectrum of pollutants. Thus, our results will benefit the future development of early warning systems for both aerosol and gaseous pollutants and toxic substances that are subject to transport processes.**

✓ **Finally, we illustrate for the first time how the use of different meteorological inputs causes differences due to planetary boundary layer height (PBLH) representation in Langrangian models. We also demonstrate how the low spatial resolution of meteorological inputs causes the omission of the sea and land breeze circulation effects on the PBLH and, as a result, the variability in radioactive tracer concentrations.**

• The authors write "Using an ensemble of meteorological inputs, this study primarily aims to investigate the seasonal and diurnal changes in the transport and surface concentration and deposition magnitude of radionuclides in the event of a potentially possible nuclear accident". I am very sensitive with radiological issues and I think they should be handled very carefully, because then can have a negative pshycholgical impact to the public. What is "potentially possible nuclear accident" supposed to mean? There is no explanation that could justify this. Why did the authors study this particular hypothetical release? Why did they not study, for example, a hypothetical release from an older reactor? For instance, several Balkan reactors (which I do not want to name, but one can easily google) from the Soviet-era have shown functionality problems during the last 10 years and could affect a more significant area (central Europe) where a larger population lives and reproduces.

✓ **While few new nuclear power plants are licensed in the Western world, and most Soviet-era stations are nearing the end-of-life decommissioning, several nuclear facilities are planned or proposed, and in the last few years are under construction or becoming operational in the Middle East/North Africa (MENA) region. The Barakah station is the latest NPP to become operational in a region**

with unique climatological conditions that were previously void of such developments and where the risk from radionuclide dispersion received little coverage in the literature, as opposed to Europe and the US.

✓ **In that regard, we also address what levels of radionuclide concentrations and deposition may affect the populated areas of Qatar in the event of a nuclear accident, a matter of significant concern due to the geopolitical situation in the region. The particular location has been selected because Barakah is the first nuclear power plant in the region, and additional ones have been planned.**

✓ **It is beyond the scope of this study to designate the causes or estimate the probability of a nuclear accident. Information about the risks of nuclear accidents is not shared by the industry and governments but needs to be taken seriously. We simulate a fictitious accident at the severity level of the Fukushima disaster but note that our results are indicative and can be scaled for any magnitude of emission from a small leak or release of radionuclides from an INES7 accident.**

✓ **The following explanation will be added to the revised manuscript: We simulate a fictitious accident at the severity level of the Fukushima disaster to compare the simulations with those produced for this accident.**

- Usually, for the assessment of transport of radionuclides and the impact of meteorological fields in transport modelling, more sophisticated state-of-the-art databases are used. I would encourage the authors to use the ETEX (Nodop, K., Connolly, R., and Girardi, F.: The field campaigns of the European Tracer Experiment (ETEX): Overview and results, Atmos. Environ., 32, 4095–4108, 1998) and ETEX-2 (https://doi.org/10.1016/j.atmosenv.2008.07.027) experiments and repeat their assessment rather that a hypothetical release that may never happen or cause the aforementioned problems (see previous comment).

   ✓ **We would like to thank you for your comment and the proposed references. Indeed, there are several studies assessing Langrangian dispersion models using controlled release experiments. As stated above, our aim is different and not directly comparable with ETEX. Repeating ETEX would not provide any information about the transport of nuclear tracers or other toxic substances in the Middle East. The current study is a contribution to the establishment of an early**

**warning system in Qatar, being the first country in the region that is planning such a system. We feel that it is important for scientists in our field (and ACP) to reach out to this region, and not only focus on Europe.**

✓ **The analyses presented in this study are based on the median of numerous simulations (1460 simulation days or 35040 simulation hours at each point) to capture diurnal and seasonal variations throughout the year and uniquely capture the uncertainty from the input meteorology. Hence, we believe that our findings related to the seasonal and diurnal changes in the transport efficiency and the concentration and deposition of radionuclides and their spatial distribution are both timely for the region of interest and relevant for scientists and decision-makers for designing early warning systems and the preparedness for potential nuclear accidents.**

- - An alternative solution for publication might be to focus on the model developments they have done, correct the manuscript and submit to GMD. This would require a detailed validation of the results, which lacks here.

  ✓ **Other than section 3.3 which concerns developments and the performance analysis of FLEXPART/FLEXPART-WRF, the major part of this study is devoted to the topics outlined above. The main focus is on the seasonal and diurnal changes in the transport and deposition of radionuclides to the region of interest (and in Qatar, subsection 3.1). We further analyze the temporal and spatial distribution of radioactive materials, the distribution of radionuclides in relation to the population density, the synoptic patterns leading to the transport of dense radioactive plumes, and the sensitivity to atmospheric turbulence. We feel that these topics are suitable and aimed toward the subject matter and audience of ACP rather than GMD.**

- In line 175, the authors are talking about a nuclear accident, but then release particles for only 24h? During the 2 worst nuclear accidents (Chernobyl and Fukushima), emissions lasted much longer, which makesa the study completely un realistic.

  ✓ **Our study is not replicating previous accidents, rather we simulate emissions over 24-hours for each day over a full year period. This, along with aggregating statistically the median output, amounts to a continuous emission over a full year**

and allows us to gather meaningful representation of the seasonal and diurnal median changes in the distribution of radioactive materials basis (in total we emit over 365 days and simulate 1460 days). The diurnal variation in the radionuclide dispersion is also considered by stratifying the simulated concentrations (Figure 2) and deposition (Figure 3) corresponding to the hourly age of the lagrangian particles. We designed this analysis (along with those shown in Figures 4 and 5) to determine what time of the day and year is associated with the higher probability of the transport of dense radionuclide plumes from a hypothetical release in the Barakah nuclear plant to the study area. To the best of our knowledge, this is the first time that such a method is implemented, and we believe that it can be used to provide important information and guide the formulation of preparedness plans.

✓ We note that in terms of emission magnitude, to translate our findings for a case in which an event with different intensities would occur, one can simply linearly scale the reported concentration/deposition risk. A realistic accident could be simulated, when it occurs, by applying our methodology in an early warning context and scaling the source strength based on available information about the accident. Similarly, by essentially simulating continuous release over a full calendar year, we can probabilistically capture the eventualities irrespective of the length of the release. Our methodology follows other studies that did not set out to determine the source term but to investigate the spatio-temporal distribution of pollutants (due to the effect of atmospheric/modeling conditions). For instance, Leadbetter et al. (2022) used a hypothetical release of 1 PBq Cs137 equivalent over 6 h at an elevation of 50 m.

- Same paragraph later mentions that "... particles are initially distributed at height levels between 100 and 300 m above the ground level over the emission point". Since we have a nuclear accident and given our previous experience with nuclear accidents, one may expect emissions at higher altitudes (see paper from Stohl's group) depending of course if there was a thermal explosion (such as in Chernobyl) or a hydrogen explosion (such as in Fukushima). Hence, one understands that a sensitivity study is also required to examine what the impact of injection altitude would be on transport. I would expect large differences on transport between emissions that occurred at 300 m and at 3 km (such as those that were calculated for the 2 major nuclear accidents in 1986 and 2011).

- ✓ **In model sensitivity studies of the emission altitude (Evangeliou et al., ACP 2013; Table 1) we note that in the case of Chernobyl, other than the first few days when the graphite core was on fire (a deprecated design), the bulk of the emissions occurred at lower altitudes. Our study is indeed based on the paper by Stohl et al. (Figs. 4, 5) for the more recent and relevant example of Fukushima. We note that in that paper the inversion over three emission layers in altitude shows that for all practical purposes, the emissions were predominantly (almost 100% for Cs137) within the 0–50 m, and 50-300 m layers.**

  - ✓ **Besides, carrying out sensitivity studies (in addition to what we have done for the turbulence schemes) causes a significant increase in the calculation load.**

- - Line 281: "Using conversion factors from Spiegelberg-Planer (2013), 131Iconc_seas_max (in a unit of Bq m-3) are converted to the maximum hourly doses from inhalation (in a unit of µSv)". This is not a proper dose-rate calculation. I would encourage the authors to calculate inhalation doses using the models presented in the WHO report for Fukushima that is the most recently updated: https://www.who.int/publications/i/item/9789241503662

- **Thank you for your suggestion. We agree with your comment and recalculated the inhalation dose based on the suggested reference and add it to the revised manuscript.**

  - ✓ **Please see the lines 292 to 300:**

    *Using the model proposed by WHO (2012) for internal dose from inhalation, $^{131}I^{intg\_conc\_seas}$ (in a unit of Bq m$^{-3}$) is converted to the effective dose from inhalation of $^{131}I$ (in a unit of µSv). The model inputs are specified for three age groups of 1-year-old infants, 10-year-old children, and adults (11 years old and up). Considering that about 90% of Qatar's population is in the adult age group (UNStats, 2020), the inhalation doses computed for this age group are discussed here (Fig. 5) and those for two others are available in the supplement (Figures S7 and S8).*

LEADBETTER, S. J., JONES, A. R. & HORT, M. C. 2022. Assessing the value meteorological ensembles add to dispersion modelling using hypothetical releases. *Atmospheric Chemistry and Physics,* 22**,** 577-596.

SAUVAGE, B., FONTAINE, A., ECKHARDT, S., AUBY, A., BOULANGER, D., PETETIN, H., PAUGAM, R., ATHIER, G., COUSIN, J.-M. & DARRAS, S. 2017. Source attribution using FLEXPART and carbon monoxide emission inventories: SOFT-IO version 1.0. *Atmospheric Chemistry and Physics,* 17**,** 15271-15292.

ZHU, C., KANAYA, Y., TAKIGAWA, M., IKEDA, K., TANIMOTO, H., TAKETANI, F., MIYAKAWA, T., KOBAYASHI, H. & PISSO, I. 2020. FLEXPART v10. 1 simulation of source contributions to Arctic black carbon. *Atmospheric Chemistry and Physics,* 20**,** 1641-1656.

Salminen-Paatero, S., Vira, J., and Paatero, J.: Measurements and modeling of airborne plutonium in Subarctic Finland between 1965 and 2011, Atmos. Chem. Phys., 20, 5759–5769, https://doi.org/10.5194/acp-20-5759-2020, 2020.

Evangeliou, N., Balkanski, Y., Cozic, A., and Møller, A. P.: Simulations of the transport and deposition of 137Cs over Europe after the Chernobyl Nuclear Power Plant accident: influence of varying emission-altitude and model horizontal and vertical resolution, Atmos. Chem. Phys., 13, 7183–7198, https://doi.org/10.5194/acp-13-7183-2013, 2013.

Stohl, A., Seibert, P., Wotawa, G., Arnold, D., Burkhart, J. F., Eckhardt, S., Tapia, C., Vargas, A., and Yasunari, T. J.: Xenon-133 and caesium-137 releases into the atmosphere from the Fukushima Dai-ichi nuclear power plant: determination of the source term, atmospheric dispersion, and deposition, Atmos. Chem. Phys., 12, 2313–2343, https://doi.org/10.5194/acp-12-2313-2012, 2012.

- A general comment is that the authors use short (24h) releases (for 365 days) and follow these for 96h. If I understood correctly, in evaluating the impact of these releases the authors look for i) the (seasonal/annual) median of the maximum concentration over each period (24h release with 96h tracking) and ii) the median of the maximum (total over 96h) deposition over each period.

✓ **Indeed, this is exactly what we have implemented. In the revised submission, according to the last comment of the first referee, instead of the maximum value of I-131 concentrations in each 96-hour simulation period ($^{131}\text{I}^{conc\_seas\_max}$), the 96-hour integration of I-131 concentrations ($^{131}\text{I}^{intg\_conc\_seas}$) is used to calculate inhalation doses (WHO, 2012). The seasonal median of recalculated inhalation doses are presented in figures 5 (for Adult), subplot 7-A (full-year analysis for adults), S7 (for infants), and S8 (for children).**

✓ **No changes were necessary in the analysis related to $^{137}\text{Cs}$ deposition while the color scale has been unified to better visualize inter-seasonal and inter-model variations of radionuclide deposition and concentrations (please see figures 5, 6, 7, S7, and S8). For clarity, instead of $^{137}\text{Cs}^{depos\_seas\_max}$, the new abbreviation $^{137}\text{Cs}^{tot\_depos\_seas}$ is applied to the seasonal median of total $^{137}\text{Cs}$ deposition.**

- However, usually the release from a nuclear accident has a longer (than 24h) duration. The authors should clearly discuss in the paper how these results may be used to understand what happens in a real case (multiple days release). For example, can this be considered a sort of median daily worst-case scenario? Could it be converted linearly in a season/yearly median worst case by considering it over a longer release period in any season or over the year?

✓ **We conducted consecutive daily simulations with release periods of 24 hours to investigate the effect of variability of atmospheric conditions with hourly/daily temporal resolution on the median distribution of airborne and deposited radionuclides. Our study is not intended to provide actual simulations of radionuclide concentration and deposition levels after a specific nuclear accident in the study area. Rather, our study aims to uniquely capture the range of diurnal and seasonal variations in the transport processes (subsection 3.1), and concentrations and deposition (subsection 3.2) of radionuclides. With our methodology, we can investigate the times of day and year when there is a certain**

**probability (risk) of radioactive materials transport and deposition in the study area, and in particular the possible population exposure. We examined the level of uncertainty in the above research questions using different meteorological inputs and different model parameterisations and codes for the entire study.**

✓ **To emphasize these points, the following paragraphs are added to the revised manuscript.**

**Added to the lines 76 to 78:**

*To the best of our knowledge, this is the first time that such a study is conducted for potential radionuclide releases in the study region, and we expect that our results can contribute to the formulation of preparedness plans.*

**Added to the lines 184 to 187:**

*We simulated a fictitious release of radioactivity at a level comparable to the Fukushima nuclear accident. However, our study does not replicate previous accidents or simulates a specific real-world case; rather we designed this analysis to determine what time of the day and year is associated with particular probabilities of transport of dense radionuclide plumes from any hypothetical release in the study area.*

**Added to the lines 319 to 322:**

*We note that one should not expect our results to reproduce the simulations of the Fukushima NPP accident. In this study, we primarily aim to determine the relative risk by highlighting variations in the spatio-temporal distribution of radionuclides in the study area due to differences in the diurnal and seasonal atmospheric processes and modeling conditions.*

Other comments

- 1) Line 201. The authors write "The relatively lower spatial resolution of CFSv2 caused a smooth distribution of its simulated air parcel ages that is close to the average of other distributions". This does not seem correct to me. For example, in fall, spring and summer (2A-all intensities) the value of CFSv2 before 25h is generally higher than all other ensemble members (therefore cannot look like an average). Moreover, "by eye" the smoothness does not seem different to me (2A-all). I suggest avoiding this statement as it is not necessary for the discussion.

✓ **Removed from the result and conclusion sections.**

- 2) Line 210, "age distribution produced by FNL-WRF was found to be more similar to the one produced by ERA5-WRF than by FNL". This is very difficult to see from figure 2A-all in my opinion, it is somewhat clear in figure 2A-high. Did you use a metric? or is this a "by eye" evaluation? 3) Similarly, to (7) and (8) above, "Although the base model used for the production of FNL, the Global Forecast System (GFS), is also the atmospheric component of CFSv2, FNL age distributions look closer to those from ERA5- and FNL-WRF". From the figure 2A-all it is very difficult for me to see these claimed similarity/difference. Perhaps you need to add a distance metric that may objectively evaluate what distributions are closer to each other.

  ✓ **In the revised manuscript we employ a metric (the maximum normalized difference) to determine the similarity between distributions of smooth density estimates of air parcel ages. The following information was added to the revised manuscript.**

  **Added to the lines 151 to 156:**

  *For the one-by-one comparison of age distributions of air parcels from ensemble members, we have used the maximum normalized difference as defined in Eq. 8. Assuming that a and b are two distributions of smooth density estimates of air parcel ages, their normalized difference is defined as the maximum value of the absolute differences of these two distributions divided by the maximum value of these two distributions. Higher values of this metric indicate greater differences in distributions (Jin and Kozhevnikov, 2011).*

$$\textbf{maximum normalized difference } = \max\big(\text{abs}(a - b)\big)/ \max\left(\max(a), \max(b)\right) \textbf{ (8)}$$

  **Added to the lines 218 to 225:**

  *The maximum normalized distance of age distributions (Fig. 3) shows larger similarity between FNL-WRF and ERA5-WRF than between the former and FNL in all seasons other than the fall (0.3, 0.19, 0.25, and 0.38 vs. 0.3, 0.32, 0.32, and 0.41 in fall, spring, summer, winter). This may to be due to the use of meteorological inputs with the same spatio-temporal resolution and a common simulation code and, consequently, similar modeling schemes for the two former members. Although the base model used for the production of FNL, the Global Forecast System (GFS), is also the atmospheric component of CFSv2, the distribution differences were found*

*to be lower between the FNL age distribution and those from ERA5- and FNL-WRF for most seasons.*

[Figure]

**Figure 3 the seasonal maximum normalized difference of smooth density estimates of air parcel ages.**

- 4) A general comment is that the discussion (lines 209-215) related to figure 2A-all (see point 7-9 above) comparing the age distributions over the whole 96h age interval seems not objective and perhaps not needed. I think that Plot 2A-all is useful for finding/pointing to specific differences that are obvious for a specific age intervals, e.g. the large peak in FNL and ERFA5-WRF in Winter at about 10hours, or e.g. what pointed out by authors in "air parcel ages are distributed in a wider range in all seasons in FNL (note the location of the first and last peaks", and afterward find the reason for the difference/similarity with a further analysis. On the other end the attempt to compare the full extension (all ages) and evaluate the overall similarity among (two or more) lines crossing repeatedly seems to me very difficult by eye (if not impossible). This comparision would need a specific metric objectively evaluating the overall distance between the lines. Concluding, in my opinion the authors should remove the discussions of 2A-all comparing curves over the whole extension or alternatively add a metric to evaluate the overall similarities/differences among the age distributions.

  - ✓ **We followed your second suggestion. We expect that the comparison metric used is sufficiently objective to preserve the discussion related to the full extension of air parcel ages.**

- 5) Please define exactly the density plotted in Figure (2.B-up) and their normalization. Obviously, particles released later in the day have a shorter travel time, i.e. particle released at 24 hours can only travel for 96h-24h=72h. What is the integral under the curves in 0-6h, 6-12h, 12-18h, 18-23h?

  ✓ **This figure (2.B-up in the first submission and 3.B-up in the revised manuscript) is similar to 3-A (2-A in the first submission), but shows the age distribution of particles stratified in four 6-hour parts of the day. For example, the upper left panel shows the age of the particles that released until 6 am on the first day of the simulations.**

  ✓ **Results show that particle ages peak between 20 and 70 hours after the release (Figure 2) and that the age of the particles leading to moderate and high radionuclide intensities (Figures 2 and S3) is less than 20 hours. Hence, we found it unnecessary to normalize particle ages. Only a very small portion of the particles has transport time in excess of 80 hours. To clarify this, the following lines were added to the text.**

  **Added to the lines 247 to 249:**

  *This panel is similar to 2-A, but shows the age distributions plotted separately for particles released in 6-hour periods of the first day of the simulations. For example, the upper left figure shows the age distribution of the particles that is released within the first 6 hours of simulations.*

  **Added to the lines 253 to 258:**

  *Because particles that are released at the end of the day have less time to travel by the end of simulation period, a sharper fall is observed in the right tail of the age distributions during the second half of the day. However, the lifetime of most simulated particles, especially of those that caused moderate (bottom row in Fig. S3-B and S4-B) and high intensities (bottom row in Fig. 2-B and S1-B), is found to be between 20 and 70 hours after release. Consequently, the difference in release time of Lagrangian particles is not significantly affected by their age spectrum.*

- 6)Line 256/ figure 3. The authors should add the formula used to define the deposition as plotted in figure 3. The current explanation (by words) lacks clarity, and the exact mathematical formulation should be added.

✓ **Equation 9 is added to clarify the way we calculated the normalized deposition values.**

**Added to the lines 272 to 278:**

*To analyze the relationship between the age composition of air parcels and the amount of $^{137}Cs$ deposition, the deposition values cumulatively summed through time steps (j) and age spectra (i) are normalized to the total amount of $^{137}Cs$ deposition simulated at each grid cell (k) at the end of each simulation run (l).*

$$^{137}Cs_{klta_{norm\_depso}} = \begin{cases} \dfrac{\sum_{i=1}^{a} \, ^{137}Cs_{klij}}{\sum_{j=1}^{96}\sum_{i=1}^{96} \, ^{137}Cs_{klij}} & if \ j = 1 \\[4mm] \dfrac{\sum_{j=1}^{t-1}\sum_{i=1}^{96} \, ^{137}Cs_{klij} + \sum_{i=1}^{a} \, ^{137}Cs_{klij}}{\sum_{j=1}^{96}\sum_{i=1}^{96} \, ^{137}Cs_{klij}} & if \ j > 1 \end{cases} \quad (9)$$

*Figure 4 shows the normalized deposition amounts ($^{137}Cs_{klta_{norm\_depso}}$) in winter when both dry and wet deposition occur in the study area.*

- Also include the definition of upper and lower bounds of the green shaded interval.

  ✓ **This is clarified in the text 289 to 290:**
  *Error bars show the 25th and 75th percentiles (the lower and upper quartiles) and the range of normalized deposition (the upper and lower extremes) within the study area.*

- Also, may you explain why the median in figure 3 occasionally decreases? Given the 30 years half-life of 137Cs, I would expect that in any grid cell the deposition increases toward its maximum at 96 hours. Therefore, the median should always increase.

  ✓ **This is clarified in the text 281 to 283:**
  *Given that the figure shows the amount of deposition across the whole study area, the decrease in the levels of accumulated deposition at the end of the simulation period pertains to the areas that are far from the source.*

- 7)Line 257, the authors write "To perform analysis related to radionuclide concentrations, the average of the simulations in the lowest four layers of the model between 5 to 100 meters has been used". Please add the mathematical definition. Are these layers evenly spaced? If not the average over the four layers should be defined accordingly to the different vertical extensions of the layers (please specify).

✓ The information is added to subsections 2.1 and 3.2.

**Added to the lines 158 to 160:**

*the thickness-weighted averages of simulations in the lowest four model levels between 5 and 100 m agl (with layer thicknesses of 5 m, 5 m, 40 m, and 50 m) are used for the spatial analysis (subsection 3.2)*

**Added to the lines 293 to 294:**

*To perform the analysis related to radionuclide concentrations, the thickness-weighted average of the simulations in the lowest four layers of the model between 5 to 100 meters has been used.*

- 8)Figure 10, S10, S11. The quality of these plots is poor.

  ✓ **All abovementioned Figures are recreated with higher quality.**

- 8.1) I think that the full year should not be overlapped with the seasons. 8.2) There is a lot of empty space on the right of the diagonal that can be used for plotting the full year separately.

  ✓ **Changed. The full-year analysis is now shown in the upper triangular of matrix plot.**

- 8.3) On the diagonal, the colored areas should be replaced with lines so that all the seasons can be clearly distinguished. What is the title of the vertical axis?

  ✓ **Changed.**

  **Added to the lines 470 to 471:**

  *The density plot (unitless) in the main diagonal of the evaluation matrix shows the relative distribution of simulations.*

Minor comments

- Eq. 2, dW_i should be dW_ j.

  ✓ **Corrected.**

- Line 100-101. "Wiener process with mean zero and variance dt", the "dt" is missing.

    ✓ **Corrected.**

- Line 108. I think that Cassiani et al (2013) should be (2015) as the reference.

    ✓ **Corrected.**

4) In table 1, add a further column indicating the deposition scheme used in FLEXPART (10.4 vs 9.02).

    ✓ **Added.**

5) Line 202-203 the phrase "could not be so great …." is unclear. Please rephrase it.

    ✓ **Corrected, please see the lines 209 to 212:**
*The significant similarity of age distributions of $^{131}I$ and $^{137}Cs$ indicates that differences in the transport characteristics of these radionuclides, such as the wet and dry deposition rate and radioactive decay, are not large enough to cause the abundance of cases where $^{131}I$ and $^{137}Cs$ particles are not present in a common grid.*

6) Line 203 what do you mean with "base concentration" ?

    ✓ **We meant low concentrations. Rephrased to** *The close dispersion of $^{131}I$ and $^{137}Cs$ concentrations, especially at low intensities …. (lines 212-213)*

7) Line 239, may you clarify what do you mean with "to the further parts of the study area…. ". In relation with the peak at high particles age in the spring.

    ✓ **Corrected. please see the lines 244 to 246:**
*Therefore, it can be concluded that regional atmospheric circulations led to the more distant transport of radionuclides to northern parts of the study area in this season than in other seasons.*

8) Figure 3, add axis titles on both the axes.

    ✓ **Done.**

9) Figure 4 and 5, add the grid spacing in the axis and explain the units of the contour lines.

    ✓ **Grid information and units are added.**

10) Line 354, "less than above thresholds" seems awkward language to me.

✓ Corrected, **please see the lines 376 to 378:**

   *The populated areas (with a density of more than 15 people per arc-second) exhibit lower expected inhalation doses and $^{137}Cs^{tot\_depos\_seas}$ less than 200 μSv and 100 kBqm−2.*

- Figure 6A and 6B, in my opinion it would be better to use a unique (for all models) color scale here.

   ✓ **Done.**

- I think that Figure S11 should be included in the main manuscript since it is discussed in many details.

   ✓ **Done. S11 is shown as Fig. 12 in the revised manuscript.**

13) Line 474, what do you mean with "iteratively" here? The models were simply run for the 365 days in the year.

   ✓ **We meant that FLEXPART with a same setting is executed iteratively for each single day of 2019.**

**References:**

JIN, D. Z. & KOZHEVNIKOV, A. A. 2011. A compact statistical model of the song syntax in Bengalese finch. *PLoS computational biology,* 7**,** e1001108.

WHO 2012. *Preliminary dose estimation from the nuclear accident after the 2011 Great East Japan Earthquake and Tsunami*, World Health Organization.

---

## Author Response (AR2)

**Editor's comments:**

1. Please address the comments from reviewer #3.

✓ **We are grateful for the valuable comments and suggestions from both yourself and Referee #3. We have taken all of the concerns into consideration and made the necessary revisions to the manuscript (indicated by line numbers in red). Our responses to the feedback are highlighted in bold text, and the updates to the manuscript are shown in bold and italicized text.**

2. You should reference and read the paper of Karion et al. (2019) (https://www.ncbi.nlm.nih.gov/pmc/articles/PMC6605086/) in which WRF-Chem and dispersion models coupled to WRF are compared.

✓ **It is considered in the lines 244-246: "*These results point out that particle dispersion modeling is primarily influenced by both the meteorological inputs and the dispersion model of choice, which is consistent with the results of Karion et al. (2019).*"**

3. A missing part in the paper is the validation of the met data, and especially the precipitation field. How realistic are they? In its present form, you list differences between FLEXPART coupled with FNL, CFSV2, FNL-WRF and ERA5-WRF. Which member has the lowest bias in wind speed and precipitation anyway? Do you improve the meteorological data by coupling FNL to WRF with a downscaling to 10km?

✓ **Thanks for your suggestion. We have added subsection 3.1 to the text. This part includes the comparison of meteorological inputs and observations. All of the above issues are addressed in this subsection. In this part we conclude that lines 256-258: "*From these results, we conclude that the downscaled datasets correlate better with observations, at the expense of an increase in error and bias values, especially for the wind speed.*"**

4. Let's assume that a nuclear accident happens in your region of interest. Which meteorological input would you use with FLEXPART anyway?

✓ **As you mentioned, the only option would be the GFS dataset if the goal is prediction. Otherwise, we have accentuated the need to an ensemble of different meteorological inputs for the dispersion modeling of radionuclides. For example, lines 316-317: "*the differences have resulted in TIDI values varying by a factor of 2 to 10 in the south of the area of interest between ensemble members. lines 517-518:***

*We attribute this to the fact that differences, however small, in the meteorological inputs that lead to cumulative deviations in the transport and concentration calculations of atmospheric pollutants.".*

5. It is worth considering to use NCEP GFS at 0.25x0.25 degree, which is one of the easiest weather model forecasts to access and would be a natural choice to predict the dispersion of a plume.

✓ **The NCEP GFS dataset is used instead of the CFSv2 dataset in the revised submission.**

6. Detailed information on your WRF simulations is missing: which PBL scheme do you use, which convection scheme, surface scheme etc.

✓ **Overview of the WRF model configuration is presented in Table 1.**

7. By using the total column, you discard, to some extent, any effects on vertical transport due to the PBL development. Actually, the capability of the model to simulate correctly the PBL is critical when estimating surface concentration or deposition. The work on the total column should be limited to the general analysis of the transport pattern. Any intercomparison between models should be removed and only the first 100 or 150m should be used.

✓ **The total column analysis is thoroughly removed. Lines 156-158: "*Thickness-weighted averages of simulated concentrations, hereafter referred to as near-surface concentrations, are calculated from concentrations within the bottom four model layers between 5 and 100 m agl (with layer thicknesses of 5 m, 5 m, 40 m, and 50 m).*"**

8. You show total deposition while there is interesting information in wet and dry depositions. Wet deposition will vary with the cloud field and precipitation in the met data.

✓ **Thanks for your suggestion. The performance analysis of the ensemble members is considered separately for dry and wet Cs-137 deposition in subsection 3.3 in lines 453-470. Please also see Figure 13.**

9. The dry deposition scheme will be directly affected by the boundary layer scheme. 417 trajectories per hour is not enough to get a good statistic at the surface. You should increase the number of trajectories by a factor of 10. You can reduce the compuation time by limiting the trajectory length (with an ageclass) to 96 hours.

✓ **We performed a test run based on the GFS dataset with a ten-fold increase in the number of particles (100000 particles in the first 24 hours of each 96-hour**

simulation period or ~ 4000 per hour). The results showed that the dry deposition simulations based on 10000 vs. 100000 particles agree well (S1). Therefore, to reduce the computational load, 10000 particles were used for all runs. Please see the lines 197-201: *"To assess if this rate is sufficient, we study the dry deposition process, which is directly influenced by the boundary layer conditions. We performed a preliminary test run with GFS data with a 10-fold increase in particles. As shown in S1, the simulated dry deposition of $^{137}$Cs and $^{131}$I and the wet deposition of $^{137}$Cs do not undergo significant changes with the increase of an order of magnitude."*

7) CFSV2 is definitively an outlier in your ensemble. As referee #3 mentioned, the CFSV2 at a 0.5x0.5 degree resolution met data should not be used within the analysis since obviously, only 3 grid cells separate the source and receptor. You should use the 0.2 x 0.2 degree version. As referee #3 mentioned: "the poorer spatial resolution of CFSv2 inputs caused the faulty separation of land and sea boundary layer process along the coastlines of Qatar. This also may lead to the suboptimal modeling of particle dispersion across the study area, especially along the coastlines"

✓ **The NCEP GFS dataset is used instead of the CFSv2 dataset in the revised submission.**

8) The downscaling exercise is difficult. In your paper, you compare the results of CFV2 to the WRF results, but they are based on completely different models. I recommend that you test your best FLEXPART-WRF set up, and go to a 4km resolution. I understand that it will be computationally hard to reduce the resolution to 4km, but you can probably do it for representative weather situations (a couple of days for each season for instance).

✓ **We have tried to implement your suggestion by downscaling FNL to the 4 and 2 km resolutions. However, in both cases, we have faced technical issues to reconcile the simulation output with the observations. Therefore, we decided not to include the 4km- and 2km sensitivity analysis in this paper, and leave that for a future study.**

✓ **However, based on your suggestion, we have added subsection 3.1 for the evaluation of the meteorological inputs. In this section, the effect of downscaling on the meteorological inputs and, subsequently on the radionuclide simulations, is examined by comparing the FNL and FNL-WRF datasets with**

**meteorological observations and with each other (before implementing different dispersion models).**

**Referee #3's comments:**

Major concerns:

1) 157: "Modeled concentrations are vertically integrated…" I see no reason what this should be good for (if not for a lack of particles in the surface layer following from too few particles being released). For any impact analysis, it is only layer (mostly surface up to ~ 150 m) concentrations (or doses) that count. Moreover, this approach makes subsections 3.2 and 3.3 incomparable. Model inter-comparison or evaluation will be biased to better outcomes because the vertical cloud or particle positions are no longer important in a total column comparison and upper layer concentrations are less impacted by tricky boundary layer processes. Total columns are mainly used in air quality studies for contrasting model values to satellite observations and this is the first time that I find total column values in a radionuclide dispersion study. The problem culminates in l. 390: "…is of importance in preparedness programs. Figure 9-A shows the frequency of occurrences (FoO) of 131I column densities…" If the authors seriously refer to column loads in the context of nuclear accident preparedness programs, there seems to be a lack of understanding.

✓ **Thank you for your suggestion. In the revised version, the weighted average of surface (0-100 m agl) radionuclide concentrations is used instead of column loads in the revised submission.**

✓ **With respect to point in l. 390, we have assumed that the high column loads coincide with the high surface values. However, column loads are not used in any of analyses in the revised version.**

2) Meteorological input data: l. 166/167: "…CFSv2 can be used to provide 6-hourly forecast inputs for FLEXPART at the spatial resolution of 0.5 degrees…" Given the scale of the study the use of this data set should be completely avoided. The emitter in UAE and the eastern boarder of the receptor area are just ~1.5° (i.e., three grid boxes) apart. It comes as no surprise that a lot of spatial gradients are lost in Figures 5 and 6. I fear that a lot of differences the authors are discussing between this member and the others are mere artefacts caused by the poor spatial and temporal resolution of the CFSv2 data. The authors even emphasize the problem of low resolution themselves

several times in the paper. The CFSv2 data should by no means be used for the purpose of forecasting in the case of an emergency.

- ✓ **The NCEP GFS dataset is used instead of the CFSv2 dataset in the revised submission.**
- ✓ **In the previous submission, we had discussed the limitations of CFSv2 for local-scale modeling in the previous submission. However, you have correctly raised the point that the use of this dataset has impacted our analysis and respective conclusions. Therefore, it is replaced by GFS dataset in the revised submission.**

Additionally, the (re-)analysis ensemble members are correlated (especially ERA5-WRF and FNL-WRF) and not well suited to quantify meteorological uncertainty.

- ✓ **In subsection 3.1 (lines 213-240) we discussed that simulations based on the ERA5-WRF and FNL-WRF datasets show only a relatively better agreement. The comparison of these meteorological inputs with observations and with each other (Figure 2), and also the comparison of the resulting simulations (Figures 12 and 13) show significant differences though. Even in the case of the FNL- and GFS-based simulations, the significant similarity between the meteorological inputs (Figure 2) does not prevent significant differences between the resulting simulations (Figures 4, 6, 7, 12, and 13). There are notable differences in the spatio-temporal distribution of radionuclide simulations based on the FNL and GFS datasets. We believe that our results highlight the need to the ensemble of different meteorological inputs for the dispersion modeling of radionuclides. For example, lines 312-317: "*The advance of TIDI above 2500 μSv to the southeast of Qatar in simulations based on FNL inputs occurs in both fall and winter, but is observed only in winter in the GFS-based simulations. TIDI values peak in the fall in both downscaled runs, with inputs from the ERA5- and FNL-WRF datasets, but the extent of high TIDI values to the southeast of Qatar in fall is much larger in the ERA5-WRF run. This is also the case when comparing the FNL-WRF and FNL-based simulations. The differences have resulted in TIDI values varying by a factor of 2 to 10 in the south of the area of interest between ensemble members*"**

3) The authors often mix integrated with maximum I-131 concentrations or integrated Cs-137 deposition with completion of C-137 deposition. It is only the maximum concentration and the completion of deposition which can be reasonably contrasted to

particle release times. Any time-integrated value naturally loses its time stamp information. Terms are correctly introduced by the authors in l. 326-328. But the statement in l. 348-349 already starts to confuse the reader. It culminates in the statement in l. 371/372, l. 523 or in the caption of Figures 7 and 8 that the integrated I-131 [muSv] concentration is converted to maximum hourly dose [muSv/h].

✓ **All above issues have been addressed. In the corrected version, it is stated that we have only checked the release time and age of particles corresponding to the maximum amount of I-131 concentrations and completion of C-137 deposition.**

4) The comparison with the results of Maurer et al. (2018) (l. 441/442, 447-450 and l. 569-571) to me demonstrates that the authors have a poor knowledge of atmospheric radionuclide dispersion modelling. Not only that Xe-133 – in contrast to I-131 or Cs-137 – is an inert tracer which undergoes no deposition (thus being easier to model) and that Maurer et al. (2018) employed atmospheric dispersion runs based on NWP (re-)analyses only (no forecast was involved), the scale of this study was completely different. Whereas the scale of the present study covers 200 or 300 km the source and the receptors in Maurer et al. (2018) are mostly several 1000 km (up to 17000 km) apart! Finally, IMS sampling times are not one hour but rather 12 to 24 hours. So, I am sorry to say, this is comparing apples with pears.

✓ **We agree with the referee's comment. In response, we have removed any comparison between the simulations of I-131 or of Cs-137 and Xe-133. In the revised version, we cite Maurer et al. (2018) only as a reference for the evaluation metrics.**

✓ **We only wanted to show that two radionuclides that are more difficult to model than Xe-133 are still within a reasonable range compared to simulations based on the reanalysis dataset (which in our study represent observations). However, as you also pointed out, we now find this comparison far from valid for several reasons.**

The abstract should cover the full paper in a balanced way. E.g., there is no word about the dose calculations.

✓ **We have revised the abstract to better represent the scope and significance of the work. The updated abstract provides a clearer and more comprehensive overview of the research aims, methods, results, and conclusions. We believe that the revised**

**abstract more accurately reflects the contributions of our study and will be of greater benefit to the readers.**

The English is sometimes poor and sentences hard to understand. Tenses are often switched arbitrarily (present tense versus past tense). Some of the minor issues below are mere suggestions, others clearly reflect a lack of care in terms of contents or wording. Sometimes authors are even contradicting themselves within the paper. It is urgently needed to increase coherence and consistency within the paper.

✓ **We have thoroughly reviewed the English language of the manuscript to ensure its clarity and coherence. Our efforts have focused on correcting grammar, spelling, punctuation, and syntax errors, as well as improving the readability of the text. To optimize the language of the MS, we will also use the Copernicus proofreading service.**

Minor issues:

- l. 11 ff.: "intensity of radionuclides" -> "(activity) concentrations of radionuclides". No proper wording. Occurs numerous times throughout the paper. Remove all occurrences of "intensity" or "intensities" in the very same context in the paper.
✓ **Done**.
- l. 12, l. 50, l. 182, l. 184: "a fictitious accident" -> "fictitious accidents". You investigated in fact 365 scenarios.
✓ **Done.**
- l. 23/24: The difference in input PBLH explains well the inter-member variations of simulated radionuclide concentrations. See major concern 2). The PBLH alone will not explain all the differences.
✓ **The entire discussion of differences in PBLH is removed.**
- l. 24/25: "Simulated concentrations were found with the same level of consistency as reported for real case studies". See major concern 4).
✓ **Corrected.**
- l. 38: "…from the Fukushima nuclear power plant accident…": Somehow a contradiction to what is said above ("…case studies of real accidents of the order of a few days are not suited to examine the impact of seasonal (atmospheric) changes on the radionuclide dispersion."). This was a real accident and the effect of East Asian northeast monsoon on radionuclide transport was evidently studied.
✓ **Here, we express the need to study a nuclear event with similar emission intensity but under different weather conditions. In fact, if the Fukushima incident had occurred under different weather conditions, the distribution of radionuclides could be very different from what was observed. This is where it becomes necessary to conduct studies similar to what we have done.**
✓ **lines 42-47**: *In addition, Long et al. (2019) studied the effects of the East Asian northeast monsoon on the transport of radionuclides from the Fukushima nuclear power plant accident to the tropical western Pacific and Southeast Asia. They found*

*that in these regions, radioactivity levels are lower than in other regions of the Northern Hemisphere, which is due to the late arrival of the radionuclide plumes carried by the monsoon circulations. That is, the dispersion of radionuclides from this accident could potentially be different under other atmospheric conditions, which are only captured by the hypothetical, iterative simulation of this event at different times of the day and year.*

- l. 40: "northern hemisphere" -> "Northern Hemisphere"
- ✓ **Done.**
- l. 48: "…transport and surface concentration and deposition…" -> "transport, surface concentration and deposition"
- ✓ **Done.**
- l. 53: "131 I concentration": The main reason for the significance of I-131 are thyroid doses, not just the high activity of I-131.
- ✓ **Modified. Please see section 3.2 (Figure 6) where I-131 concentrations are converted to the thyroid internal dose from inhalation (TIDI, in units of μSv).**
- l. 55 and l. 86: Please add "Pisso et al., 2019" to the references.
- ✓ **Done.**
- l. 63: "lack of accuracy". Please specify. With regard to the internal modelling time step?
- ✓ **It is reworded to clarify. lines 67-69:** *In some cases, the particles may not remain well-mixed during simulation (Brioude et al., 2013). This is mainly due to the treatment of the stochastic motion of the particles and/or the mass balance of vertical velocity with the horizontal winds.*
- l. 68: "perturbations". Please specify and/or provide a reference.
- ✓ **All three approaches are discussed by Galmarini et al. (2004). In the revised version, this reference precedes the approaches.**
- l. 69: "suite of different meteorological models": Difference in practice may be limited due to NWP models being similar to each other and thus an ensemble can easily give an incomplete picture of meteorological uncertainty. See for example your ERA5-WFR versus FNL-WRF inputs.
- ✓ **Although the GFS- and FNL-based (and also FNL-WRF- and ERA5-WRF-based) simulations have the higher agreement due to the similarities between their meteorological inputs. But significant differences are also reported in the resulting spatio-temporal distribution of simulations. For example,** *lines 312-317: The advance of TIDI above 2500 μSv to the southeast of Qatar in simulations based on FNL inputs occurs in both fall and winter, but is observed only in winter in the GFS-based simulations. TIDI values peak in the fall in both downscaled runs, with inputs from the ERA5- and FNL-WRF datasets, but the extent of high TIDI values to the southeast of Qatar in fall is much larger in the ERA5-WRF run. This is also the case when comparing the FNL-WRF and FNL-based simulations. The differences have resulted in TIDI values varying by a factor of 2 to 10 in the south of the area of interest between ensemble members.*
- ✓ l. 71-73 "…is compared against the (re)analysis members. (Re)analysis-based simulations are expected to be closer to (unavailable in a real-world scenario) actual values than forecast-based ones (Leadbetter et al., 2022)." -> "…is compared against (re)analysis members. (Re)analysis-based simulations (unavailable in a real-world

scenario) are expected to be closer to actual values than forecast-based ones (Leadbetter et al., 2022)."

✓ **Done.**

- l. 96/97: Please check the suitability of references. Tipka et al. describes the preprocessing of ECMWF fields before being ingested into FLEXPART, the other two papers deal with convection. But likely not removal processes (decay and deposition).

✓ **Relevant references are added in lines 101 and 102 "computing various removal processes (Stohl et al., 2005, Grythe et al., 2017)"**

- l. 113: "…by (Hanna, 1982)." -> "…by Hanna (1982)."

✓ **Done.**

- l. 114: "…method as Maryon (1998) is followed." -> "…method as in Maryon (1998) is followed."

✓ **Done.**

- l. 115: "In dispersion modeling…" -> "In dispersion modeling of radionuclides…"

✓ **Done.**

- l. 118: "…from already calculated the radionuclide…" -> "…from the radionuclide…"

✓ **Done.**

- l. 123: "…the Weather Research and Forecasting (WRF)…" -> "…the Weather Research and Forecasting (WRF) model…"

✓ **Done.**

- l. 126 + 127: "cloudy pixels" -> "cloudy grid cells"

✓ **Done.**

- l. 130: "…any grid cells beneath these grid cells…" Do you mean beneath a value of 80% or beneath in terms of altitude?

✓ **Corrected. please see the lines 132-137 *"In previous versions, including version 9.0.2 used in the development of FLEXPART-WRF 3.2, in-cloud grid cells are defined as those with precipitation and relative humidity above 80%. The grid cells below the in-cloud grid cells up to the surface are defined as below-cloud grid cells (Seibert and Arnold, 2013, Pisso et al., 2019). In recent updates to the FLEXPART's source code, the above threshold has been modified using the 3D cloud water mixing ratio ($q_c$) fields. The threshold of $q_c > 0$ ($q_c = 0$) now identifies grid cells within the cloud (below the cloud) (Pisso et al., 2019)."***

- l. 131: "…cloud water mixing ratio…" This field is now used to distinguish between below- and in-cloud grid cells. Please state this explicitly.

✓ **Corrected. please see the above answer.**

- l. 141: "Eq7" –> "Eq. 7"

✓ **Done.**

- l. 148: "…estimate the transport…" -> "…estimate the temporal characteristics of transport…"

✓ **Done.**

- l. 150: "…is added to the history output grids that have a horizontal resolution of 10 km…" -> "…is added to the output grid that has a horizontal resolution of 10 km…"

✓ **Done.**

- l. 152/153 & caption of Fig. 3: "smooth density estimates". In how far smooth? Were the distributions smoothed?

- ✓ **The intensity of the smoothing is controlled by the kernel bandwidth. In our study, it is determined by using the method called Scott's rule: n\*\*(-1./(d+4)), where n is the number of data points and d is the number of dimensions.**
- • l. 153: "normalized difference" -> "maximum normalized difference"
- ✓ **Done.**
- • l. 170/171: "…reanalysis data that covers from January 1, 1950, to nearly the present. They are produced at a spatial resolution of about 31 km at hourly time steps." -> "…reanalysis data that covers January 1, 1950, to nearly the present. They are produced at a spatial resolution of about 0.25° at hourly time steps.
- ✓ **Done.**
- • l. 175/176: "A single simulation code is built for each meteorological dataset to be ingested by FLEXPART…" I do not really understand why. In FLEXPART 10.4 there is even only one executable for both, ECMWF and NCEP data. So I wonder why there should be different codes for two NCEP data sets. Finally, authors are contradicting themselves in l. 303 ("same simulation code").
- ✓ **We could not run FLEXPART with input from the FNL and CFSv2 datasets using the same executable. We had to compile two different executables files. However, as you said, the simulation code is the same. In the revised manuscript, the term "executable" is removed and we use only "the simulation code" throughout the manuscript. The paragraph above is deleted.**
- • Table 1 needs to be improved. Use capitals consistently in the header, e.g., "Temporal resolution". First line with entries: Better remove "x". Second line with entries: Add "0.25°" and "3-hourly" for FNL. Better remove "x".
- ✓ **Done.**
- • l. 184: "May 2011" - > "March 2011"
- ✓ **Done.**
- • l. 193: "This experiment has been performed…" -> "This experiment was performed…"
- ✓ **Removed.**
- • l. 194/195: "For the diurnal and seasonal stratification of simulations…" –> "For stating particle ages related to simulations…"
- ✓ **Removed.**
- • Caption of Figure. 1: "Figure 1 A is the study area embracing the B-NPP (red square) and the state of Qatar. The base map and overlaying information are taken from Google Earth. B is the schematic illustration of the LPDM simulation cycle." -> "Figure 1. A: Study area embracing the B-NPP (red square) and the state of Qatar. The base map and overlaying information are taken from Google Earth. B: Schematic illustration of the LPDM simulation cycle." I suggest a similar style for all the figures. Use full stops and colons accordingly.
- ✓ **Done. This correction is applied to all figures with a similar caption structure.**
- • l. 205: "…since it resides mostly in the gaseous phase and has a short half-life of 8 days" This is not true according to my knowledge. The best assumption is a 50:50 partition between gaseous and particulate iodine. Again, the thyroid doses are an important aspect of iodine.
- ✓ **Based on data by the "Ring of Five (Ro5)", an informal network of European national authorities (which comprises more than 150 sampling systems of high**

**volume samplers and some with activated coal traps), the average gaseous/total ratio for 131I is 77.2 ± 13.6 % (Masson et al., 2011). The US Environmental Protection Agency (EPA) RadNet station measurements detected 81 % of the ambient 131I in the gas and 19 % in the particle phase (Ten Hoeve and Jacobson, 2012). The average of 71 ± 11 % reported from the Fukushima site from 22 March to 4 April 2011 (Stoehlker et al., 2011). Therefore, we decided to consider only the gaseous phase of I-131, especially the Cs-137 can represent particulate pollutants.**

- l. 208: "…season and time of day in which…" -> …season and time of day in/at which…"

✓ **Removed.**

- Figures 2 and S1: "FNL-WRF" -> "Forecast". Are times in Fig. 2B/S1B UTC-times? Y-axis: "all intensity" -> "all loads". Caption of Figure 2: "Figure 2 A: the smooth density estimates of air parcel ages corresponding to all intensities of 131I column densities (top row) and of those above the 66th percentile (bottom row). B: the same as A, but for four times of the day." -> "Figure 2. A: Density estimates of air parcel ages corresponding to all 131I column loads (top row) and of those above the 66th percentile (bottom row). B: The same as A, but for four times of the day."

✓ **Done. All day times are converted to local time in the revised version.**

- l. 210-212: "…differences in the transport characteristics of these radionuclides, such as the wet and dry deposition rate and radioactive decay, are not large enough to cause the abundance of cases where 131I and 137Cs particles are not present in a common grid." This is even not to be expected, because removal process have no influence on particle positions in LPDM. Just on particle masses. However, the transport for gases and particulates (undergo gravitational settling) will be different to some extent.

- **Corrected. Lines 243-244:** *the high degree of similarity between the age distributions of 131I and 137Cs is due to the fact that the removal process in LPDMs only affects mass concentrations and not particle positions.*

- Figure S2: "…lines are corresponding kernel density estimates of counts." Completely obscure to me. Needs (sufficient) explanation in the text.

✓ **The explanation is added to the caption of S2 (S7 in the revised version)** *"The distribution of the ratio of 137Cs wet deposition to total 137Cs deposition for each ensemble member (colors). The counts of the ratios are smoothed with a Gaussian kernel. The counts on the y-axis are on a logarithmic scale."*

- l. 216: "All ensemble members in all seasons simulated an abrupt increase…" I think this is not true for summer.

- **This conclusion holds true in all seasons after column loads are replaced by near-surface concentrations.**

- l. 225: "In addition to having finer spatial resolution than CFSv2, FNL assimilates observations like ERA5." Also CFSv2 needs to assimilate observations at some point, i.e., at the analysis time step based on which the forecast is made.

✓ **Due to replacement of the CFSv2 datasets by GFS dataset, the statement is removed.**

- l. 226: "distributed in a wider range" Did you check the significance of this feature?

✓ **We explained this feature in more detail in lines 345-355. We believe that further elaboration of this feature is beyond the scope of this paper.**

- l. 226/227: "…in FNL…" -> "in FNL-based FLEXPART simulations" or "FNL simulated…" -> "FLEXPART-FNL simulated…" FNL does not yield the output directly. It needs FLEXPART in addition. Numerous analogous formulations throughout the paper. Please adapt them as outlined.
- **Corrected throughout the text.**
- l. 228 + l. 357: "…air parcels reaching further receptors…" -> "…air parcels reaching receptors further away…"
- ✓ **Done.**
- l. 232: "column (mass) densities" (or density). Please note that this is not a proper term. Occurs numerous times in the paper. It has to be "column load". If you vertically integrate concentrations [Bq/m3] you will end up with column loads [Bq/m2]. However, as stated above, column loads are not used in radionuclide studies.
- ✓ **This correction does not apply to the revised submission because the weighted average of surface (0-100 m agl) radionuclide concentrations replaced the column loads.**
- l. 242/243: "For low intensities of both radionuclides, however, the age distributions are almost the same as that seen in all intensities. The peak of newly arriving air parcels in all intensities occurs earlier in spring than in other seasons." -> "For low column loads of both radionuclides, however, the age distributions are almost the same as that seen for all column loads. The peak of fast arriving air parcels for all column loads occurs earlier in spring than in other seasons." I would rather say spring, winter and to some degree also fall behave very similar in terms of fast arriving particles.
- **This statement is removed because all of the above patterns are not seen in the distribution of particle ages corresponding to different levels of near-surface concentration, which replace column loads in the revised version.**
- l.244: "…in low and all intensity column densities…" -> "…for low and all column loads…"
- ✓ **Removed.**
- l. 245: "…led to the more distant transport of radionuclides to northern parts of the study area…" Can this really be concluded that easily? See the options stated at the end of the above paper paragraph.
- ✓ **This statement is removed. Please see the response to your point about l. 242/243.**
- l. 249: "…the age distribution of the particles that is released…" -> "…the age distribution of the particles that are released…"
- ✓ **Done.**
- l. 250/251: "…the number of long-lived air parcels (including all intensities of concentrations) increased in all members…" -> "…the number of long-lived air parcels (including all levels of column loads) increased for all members…" Anyway, not true for Cs and FLXPART-FNL (Figure S1).
- ✓ **In order to shorten the article (at the same time adding section 3.1 related to the comparison of meteorological inputs with observations) and also to avoid unnecessary discussions, we have removed the discussion of the effect of the release time of particles on their age distribution. However, the effect of particle release time on the intensity of radionuclide concentration and deposition are discussed in Section 3.2 (Figures 6, 7 and 8).**
- l. 253: "longer Lagrangian particle ages" -> "higher Lagrangian particle ages"

- ✓ **Please see the response to your point about l. 250/251"**
- • l. 254: "less time to travel by the end of simulation period" -> "less time to travel until the end of simulation period"
- ✓ **Please see the response to your point about l. 250/251.**
- • l. 257: "…between 20 and 70 hours after release…" I would rather say "…up to 40 or 50 hours after the release…" for moderate and high column loads."
- ✓ **Please see the response to your point about l. 250/251"**
- • l. 257/258: "Consequently, the difference in release time of Lagrangian particles is not significantly affected by their age spectrum." Sentence makes no sense. If, then the other way round.
- ✓ **Please see the response to your point about l. 250/251"**
- • l. 258/259: "…FNL have simulated a larger number of shorter-aged air parcels…" -> …FLEXPART-FNL simulated a larger number of shorter-lived air parcels…"
- ✓ **Please see the response to your point about l. 250/251".**
- • l. 261/262: "Like seasonal distribution, the diurnal variations of air parcel ages for high concentrations are very similar in all members." -> "Like for seasonal distributions, the diurnal variations of air parcel ages for high column loads are very similar for all members."
- ✓ **Please see the response to your point about l. 250/251".**
- • l. 276: This equation is quite obscure to me. If j=1 (first simulation time step) "a" can only be equal to one (j=a=1). There cannot be particles older than one hour at this stage of the simulation. For j >1, e.g. j=50, i can only range between 26 and 50 and cannot adopt a value equal to 1 in the nominator given that particles were released over the first 24 hours of the simulation.
- ✓ **Thanks for your comment. We have made changes and corrections To Eq. 9 to include all possible conditions leading to CS deposition equal to zero. Please see lines 276-284**

**"To analyze the relationship between the age composition of air parcels and the amount of $^{137}$Cs deposition, the deposition values cumulatively aggregated across time steps (j) and age spectra (i) are normalized to the total amount of $^{137}$Cs deposition in each grid cell (k) at the end of each simulation run (l).**

$$^{137}Cs_{klna_{norm\_depso}} = \begin{cases} \dfrac{^{137}Cs_{kl(n-1)n}}{\sum_{j=1}^{96}\sum_{i=1}^{95} {^{137}Cs_{klij}}} & if\ n = 2 \\[4mm] \dfrac{\sum_{j=1}^{n-1}\sum_{i=1}^{j-1} {^{137}Cs_{klij}} + \sum_{i=1}^{a} {^{137}Cs_{klin}}}{\sum_{j=1}^{96}\sum_{i=1}^{95} {^{137}Cs_{klij}}} & if\ n > 2 \end{cases} \quad (9)$$

**where n is the time step with a maximum of 96 (the last time step) and $a$ is the given particle age with a maximum of n-1. $^{137}Cs_{klij} = 0$ in two conditions (1) $n \geq 26$ if $i \in [1, n-25]$ and (2) $i \geq j$."**

- l. 277: "…in winter when both dry and wet deposition occur in the study area. We found very similar results in other seasons (Fig. S5)." Something got quite confused here. Comparing Figure S5 with Figure 4 it is easy to see that, e.g. results for FLEXPART-FNL, are not identical for season winter.

✓ **Regarding "…in winter when both dry and wet deposition occur in the study area", this shows why we chose the simulations in winter to discuss in the main text.**

✓ **We did not claim they are identical but similar: please see lines 286-287: "*Similar deposition patterns are obtained for other seasons (S6). As shown in S7, the main reason for the small difference between the seasonal deposition patterns is the lack of precipitation and subsequent wet deposition in the region.*"**

- I guess Figure 4 in fact depicts the full year. I also wonder about similar results in other seasons given that wet deposition will hardly occur in summer in the study area.

✓ **Corrected. Sorry for the mistake. It was an old figure left over from the first draft.**

- l. 277/278: "As expected, the values of the 137Cs deposition increase cumulatively with the time after the accident." This indeed should be the case. But this is not what can be seen in Figure 4. There is no steady increase in, e.g., the median. Consequently, the statement in the next line, i.e., "deposition (the median of normalized deposition > 0.8) happens within 80 hours after the assumed accident" is misleading to me. According to my understanding Figure 4 probably says that 80% of the deposition is accomplished by particles up to an age of 80 hours integrated over all possible time steps.

✓ **Regarding There is no steady increase in, e.g., the median, this represents deposition variations in the areas far from the source.**

✓ **lines 288-290: "*Although the spatial pattern of the deposition varies considerably, as indicated by the range of quartiles, the median of the normalized deposition shows that about 80 percent of the deposition occurs within 80 hours after an accident. The cumulative deposition at the end of the simulation period is mostly in the areas farthest from the source.*"**

- Caption of Figure 4: "This plot (S5) shows simulations in winter (other seasons)." Weird sentence in this context.

✓ **Corrected. lines 285-286: "*Figure 5 shows the normalized deposition amounts ($^{137}Cs_{klna_{norm\_depso}}$) in winter, when both dry and wet deposition occur in the study area. Similar deposition patterns are obtained for other seasons (S6).*"**

- l. 292: "…the seasonal and diurnal changes…" -> "…the spatio-seasonal distribution…"

✓ **Modified.**

- **Figure 5: Too busy with regard to release time isolines. I think they should be thinned out, local minima and maxima be removed, respectively. Please also state the unit, i.e., muSv.**

✓ **Corrected.**

- l. 296: "131I_intg_conc_seas": Needs to be explained when first introduced.

✓ **Done**.

- l. 297: "The model inputs…" -> "Dose coefficients…"?

- ✓ **Corrected. we meant "The conversion coefficients". Please see the lines 301-303:** *"Since approximately 90% of the population of Qatar is in the adult age group (UNStats, 2020), TIDI is specifically calculated for this age group using the coefficients defined by WHO (2012) (Fig. 6)."*

- l. 301: "…in most parts of study area than the other members." -> "…in most parts of the study area compared to the other members."

- ✓ **Due to the replacement of column loads by near-surface concentrations, this statement is removed.**

- l. 306: "…over emission point in UAE and, with less intensity, within the study area…" -> "…over the emission point in UAE and, with lower heights, within the study area…" What is the reason for this discrepancy? For which time (noon?) is Figure S9 valid?

- **Due to the replacement of CSFv2 by GFS, this statement is removed.**

- l.307: "dilution" -> "vertical dilution"

- ✓ **Removed.**

- l.308: "process" -> "processes"

- ✓ **Removed.**

- l.311: "cold period" Rather mostly fall instead of winter.

- ✓ **Corrected. See lines 305-308: "*Our ensemble simulations show that the TIDI values above 2500 µSv occur frequently in the cold period of the year, especially in fall, in the simulations of all members. This may be due in part to the lower (higher) PBLH in the cold (warm) seasons and to the synoptic conditions."***

- l. 314: "exceptional transport" Why is this feature not present in the both WRF simulations?

- ✓ **This pattern is only seen in FNL-based simulations. Therefore, I think you mean why it is not seen in the simulations based on GFS dataset. If so, we believe that "*This may be caused by a rare atmospheric circulation in summer, discussed later, and it may change as the modeling period is extended*" (lines 308-309).**

- l. 315-319: Comparison to the Fukushima accident is too vague. How do you define "adjacent"? If I am not mistaken the paper shows doses over 96 hours for iodine only. But the paper cited evidently provides integrated doses for three months and three nuclides.

- ✓ **We discuss the thyroid internal dose from inhalation (TIDI) in the revised version. Therefore, the above comparison is removed**. Lines 303-305: "*As reference values for comparison, the total TIDI values, collectively calculated for 15 studied radionuclides for the adult age group, were found to be between about 2,000 and 50,000 µSv in the first year following the Fukushima accident in areas close to the power plant (see Table 4 in WHO (2012))."*

- l. 324/325: "For example, the simulations of FNL in southern Qatar in fall are more than twice that of FNL-WRF (more details in the subsection 3.3)." -> "For example, the simulations based on FNL in southern Qatar in fall result in more than twice the dose compared to that based on FNL-WRF (more details in the subsection 3.3)."

- ✓ **Due to the replacement of column loads by near-surface concentrations, the discussion is modified.**

- l. 328-329: "To identify the highest possible level of pollution at each point, regardless of its frequency, local maxima are calculated only from non-zero intensities." Sentence is completely obscure to me.

- ✓ **Modified. lines 318-321:** "*To investigate the influence of the particle release time on the radionuclide dispersion, the seasonal median of the particle release time (hours in local time (LT)) coinciding with the maximum concentration of 131I and the completion of 137Cs deposition is considered (contours in Figures 6 and 7). The results show that the highest 131I concentrations coincide with particles released between 9 a.m. and 2 p.m. LT in most parts of the study area*"

- • l. 330: "is caused" -> "are caused"
- ✓ **Done**.

- • Figure 6 and 7B: Needs to be redone with units converted to kBq/m2, critical thresholds (40 and 90 kBq/m2, see text) clearly distinguishable and probably replacing the whitish part of the color scale.
- ✓ **Done**. **Please see Figure 7.**

- • l. 335-343: As a result of the deficiencies of Figure 6, the discussion is not really traceable looking at the figure. Specifically, 40 and 90 kBq/m2 cannot be distinguished (both thresholds fall within the withish part of the color scale).
- ✓ **Corrected. Due to the replacement of CSFv2 by GFS, 90 kBq/m2 threshold (found in CFSv2-based simulations) is removed.**

- • l. 346: "…the higher CFSv2 137Cstot_depos_seas in winter…" It is not really higher, but rather affects a broader area.
- ✓ **Due to the replacement of CSFv2 by GFS, this statement is not considered in the revised version.**

- • l. 362 + caption of Figures 5 & 7: "…particle ages simultaneous…" -> "…particle ages occurring simultaneously…"
- ✓ **Captions are rephrased**.

- • Figure 8: Convert Bq/m2 to kBq/m2.
- ✓ **Done**.

- • l. 375: "…extremely high levels of inhalation doses (higher than 200 μSv) and of 137Cstot_depos_seas (higher than 100 kBqm−2)…" Please provide a reference for these – according to your opinion - "extremely high" thresholds.
- ✓ **It is just a comparative conclusion about the simulations in the study area. Therefore, we believe it does not need a reference. We have already provided a reference for the threshold (40 kBqm-2) used for identifying contaminated areas.**
- ✓ **Please also note that inhalation doses are replaced by TIDI that changes the critical levels in the discussion above. For example, "…. *the extremely high levels of TIDI (greater than 10,000 μSv)….*"**

- • l. 379: "…131Iintg_conc_seas inhalation doses…" -> "…131Iintg_conc_seas related inhalation doses…"
- ✓ **Corrected. we refer to TIDI in the revised version.**

- • l. 379/380: "…there are no pronounced seasonal and inter-member differences in 137Cstot_depos_seas at different population levels." This is hard to believe looking at Figure 6.
- ✓ **Corrected. Lines 369-372:** "*Due to the exceptional weather pattern that occurs in the simulations based on the FNL dataset in the eastern part of Qatar (where the most densely populated areas are located) in summer, these simulations cause the highest values of TIDI in densely populated areas. Otherwise, all ensemble members simulate the highest TIDI during the cold seasons. The highest $^{137}Cs^{tot\_depos\_seas}$ in the*

*same areas are observed based on ERA5 and FNL-WRF in the cold period of the year."*

- l. 391/392: "…north of Qatar…" -> "…in northern Qatar…"
- ✓ **Done**.
- l. 394: "…15% of extreme events have reached receptors in Qatar." -> "…15% of extreme events occur in Qatar."
- ✓ **Done**.
- l. 398: "has caused"-> "causes"
- ✓ **Done**.
- l. 400/401: "…occurs mainly in the late winter-early spring period simultaneous with the southward movement of westerlies and the eastward movement of the Saudi Arabian subtropical high pressure." -> "…occurs mainly in the late winter-early spring period simultaneously with the southward movement of westerlies and the eastward movement of the Saudi Arabian subtropical high pressure system."
- ✓ **Done**.
- l. 410: "As shown in Figure 10-A, the median of simulated column densities…" -> As shown in Figure 10-A for FLEXPART-ERA5-WRF, the median of simulated column loads…"
- ✓ **Due to the replacement of the column loads by near-surface concentrations, the discussion is modified**.
- l. 411: "There is no comparable change in other seasons that is reflected in a minute increase in the full-year distribution of…" -> "There is no comparable change in other seasons which is reflected in a minute increase in the full-year distribution of I-131 load for…"
- ✓ **Due to the replacement of column loads by near-surface concentrations, the discussion (lines 404-414) is modified**.
- l. 413-414: "The upper quartile of 137Cs deposition with STM in winter is around 25% higher than those simulations with GTM in the same period." -> "The upper quartile of 137Cs deposition related to STM in winter is around 25% higher than that related to GTM in the same period.
- ✓ **Corrected (lines 404-414)**.
- l. 415: "The implementation of STM in FNL-WRF…" -> "The implementation of STM in FLEXPART coupled with FNL-WRF…"
- ✓ **The discussion (lines 404-414) is modified**.
- l. 417: "mainly in fall": What is the synoptic pattern related to the feature being pronounced in fall?
- ✓ **In this part, we discuss the effect of the turbulence scheme of choice a local scale. Therefore, the discussion of synoptic patterns seems irrelevant. In addition, the discussion (lines 404-414) is modified due to the replacement of column loads by near-surface concentrations.**
- Figure 10: Please change the y-axis units to kBq/m2 (you can probably stay with Bq/m3 for concentrations if you remove the vertical integration) and introduce scientific notation.
- ✓ **Done. Please see Figure 11.**

- l. 434: "…from the six International Monitoring System (IMS) stations…" -> …from six International Monitoring System (IMS) stations…" There are more than 70 IMS radionculide stations in total!
✓ **Done**.
- l. 436: "…normalized by the sum of the two means…" -> "…normalized by the sum of the simulation and measurement means…"
✓ **Done**.
- Figure 11: I wonder about the spurious beams in the regression analysis. Especially about the blue ones in the regression analysis for the full year (yellow data points). I would expect simple lines. Moreover, the seasonal subplots can be skipped in my opinion. In the present layout they are too busy with the red points for fall covering much of the remaining data points. Additionally, they are hardly discussed at all in the text (just l. 443 and 444)
- **All season-related discussion is removed to address the above issues and to make room for the discussion of wet deposition.**
- l. 446 ff.: "FB5" -> "F5" Occurs numerous times in the subsequent paragraphs.
✓ **Done**.
- l.452: "…the normalized RMSE which is less sensitive to extreme values…" Sorry, I do not understand. Large deviations will be even enforced due to the quadratic term involved. Also, the NMSE is not equal to the normalized RMSE. Apart from normalization the square root is not applied for NMSE.
✓ **Thanks for the point. In fact, in our analysis in the previous submission, we used the normalized RMSE. But after your comment, we checked the reference and now use the correct one.**
✓ **Due to the replacement of the column loads by the near-surface concentration and the replacement of CFSv2 by GFS, this discussion is no longer valid.**
- l. 453: "(2)" -> "(2.6)"
✓ **Please see the answer above**.
- l. 454/454: "Feeding downscaled FNL inputs into FLEXPART-WRF (FNL-WRF) increased the correlation of ERA5-WRF and FNL-WRF to 0.7." - > "Feeding downscaled FNL and ERA5 inputs into FLEXPART-WRF (FNL-WRF and ERA5-WRF) increased the correlation to 0.7.
✓ **Rephrased**.
- l. 456/457: "…indicating that the downscaling of inputs in FNL-WRF did not have much effect on the association of their simulations." Comparing this statement with Figure 5 (where there is a considerable difference between FNL and FLN-WRF based simulations) the inconsistency introduced by using column loads and surface concentrations (doses) alternately becomes very evident.
✓ **We discussed here just the relative association (correlation) of simulations (not their absolute difference). Anyway, the column loads are replaced by the near-surface concentrations.**
- l. 458-460: "This suggests that the downscaling of similar (FNL) and different (FNL and ERA-5) meteorological datasets increased and decreased the absolute differences between resulting simulations, respectively." This sentence makes no sense to me. FNL is similar to what? Overall it is evident from the statistics and does not come as a surprise that simulations based on ERA5-WRF and FNL-WRF are most closely related.

- **Rephrased. lines 441-444:** "*On the other hand, the downscaling of the inputs and the application of the same simulation code yield higher agreement of the simulations based on the FNL-WRF and ERA5-WRF datasets than to those based on the FNL and FNL-WRF datasets (r=0.7 vs. 0.67, FB=0.01 vs. -0.93, F5=69.83% vs. 53.3%, RMSE=36012.82 Bq/m3 vs. 113888.80 Bq/m3, and NMSE=2.74 vs. 10.08).*"
- l. 460/461 and caption of Figure 11: "…relative distribution of simulations…" -> "…relative column load distribution of simulations…"
- ✓ **Modified**.
- l. 461/462: "All distributions here depicted the higher frequency of low 131I column densities in spring than in other seasons (as also seen in Figure 2-A)." Figure 2-A does not display the frequency of column loads but rather that of particle ages! What is reason for this exceptional feature? Why does it not occur in summer as well?
- ✓ **Regarding old results, this may be due to synoptic patterns in spring that transport radionuclides to areas further away from the source.**
- ✓ **As requested, all season-related discussion is removed.**
- l. 463: "While the RMSEs have increased tangibly…" I would suggest checking this result. I would have expected it rather the other way round because simulated I-131 levels will overall be higher compared to Cs-137 levels due to the larger source term. Thus, absolute differences between modelled values are prone to be larger for I-131.
- ✓ **Corrected. We apologize for this error. The I-131simulations seem to be drawn in the unit of ng/m3 in the previous submission. They are in the correct unit (Bq/m3) in the revised submission.**
- l. 474: "The other statistics…" -> "The statistics…"
- ✓ **Removed.**
- l. 496: "We quantified meteorological uncertainties by producing an ensemble model…" You can hardly quantify the meteorological uncertainties based on three re-analysis members which are correlated (FNL – FNL-WRF, but above all ERA5-WRF – FNL-WRF). The ECMWF, e.g., uses 50 (!) non-redundant ensemble members for uncertainty quantification. Running FLEXPART based on these (or at least a fraction of these) members would deserve the term "uncertainty quantification". See major concern 2).
- ✓ **We discussed that simulations based on the ERA5-WRF and FNL-WRF datasets are only in a relatively better agreement. The comparison of these meteorological inputs with observations and with each other (Figure 2), and also the comparison of the resulting simulations (Figures 12 and 13) show significant differences. Even in the case of the FNL- and GFS-based simulations, the significant similarity between the inputs (Figure 2) does not prevent significant differences between the resulting simulations (Figures 4, 6, 7, 12, and 13). For example, lines 316-317: "*the differences have resulted in TIDI values varying by a factor of 2 to 10 in the south of the area of interest between ensemble members. Lines 517-518: We attribute this to the fact that differences, however small, in the meteorological inputs that lead to cumulative deviations in the transport and concentration calculations of atmospheric pollutants.*".**
- l. 502: "…were also examined concerning the population density…" -> "…were also examined in relation to the population density…"
- ✓ **Done.**

- l. 513 "in spring" I would say in fall too.
- ✓ **Due to the replacement of column loads by near-surface concentration, this conclusion is no longer valid.**
- l. 527: "stronger" -> "larger"
- ✓ **Done.**
- l. 535: "We found…" -> "We suspect…"
- ✓ **The discussion related to PBLH is removed.**
- l. 541: "in winter" It looks like (Figure 6) this pattern is even more pronounced in fall.
- ✓ **Corrected. Lines 520-523: "*The largest expansion of areas with $^{137}Cs^{tot\_depos\_seas}$ greater than 40 kBqm$^{-2}$were found in the simulations based on the ERA5-WRF and FNL-WRF datasets in winter. The highest levels of $^{137}Cs^{tot\_depos\_seas}$ (above 300 kBqm$^{-2}$) were found in the simulations based on the ERA5-WRF dataset in the southeastern corner of Qatar in the fall.*"**
- l. 550: "…in which south/southeast winds transport…" -> "in which south/southeast winds in the cold season transport…"
- ✓ **This is not necessary, as "winter" is mentioned in two lines up (line 531).**
- l. 556/557: "…decreased the median of simulated 131I concentrations…" Only in fall.
- ✓ **After the replacement of total column by near-surface concentrations results changed as follow: Lines 537-542** "*The implementation of the Skewed Turbulence Model (STM) instead of the Gaussian Turbulence Model (GTM) increased the occurrence of high levels of $^{131}I$ concentrations and $^{137}Cs$ deposition. For example, the quartiles of simulations of $^{131}I$ concentration and $^{137}Cs$ deposition based on the ERA5-WRF dataset increased by 21% and 12%, respectively. According to Pisso et al. (2019), this can be interpreted as the enhancement of concentrations and deposition in the areas around the source under skewed turbulence conditions*"
- l. 561: "In general, CFSv2 simulations of 137Cs and 131I column density are most highly correlated with FNL…" -> "CFSv2 simulations of 137Cs and 131I column loads are most highly correlated with FNL…" Mind that FNL – FNL-WRF correlation is highest.
- ✓ **Due to the replacement of CFSv2 by GFS, this conclusion is removed.**
- l. 563: "FB = 0.02" -> "FB = -0.02"
- ✓ **Removed.**
- l. 560-575: Please avoid stating numerous statistical scores in the conclusion sections. There can be two or three of them but the conclusion section should contain take-away-messages rather than multiple numbers.
- ✓ **Done.**
- l. 573: "RMSE=14.6 x 10^8 and 2946.3 x 10^7"? Exponents?
- ✓ **Corrected. Please see also answers for l. 561 and l. 463.**
- l. 576: "…simulations from FNL and FNL-WRF with identical meteorological inputs." FNL and FNL-WRF are not "identical", but of course correlated to some degree. The authors should decide at some point in the text whether their ensemble input quantifies meteorological uncertainties (see l. 496) or is identical which makes the use of an ensemble obsolete.
- ✓ **In the corrected version, in several parts, we concluded that the simulations of the ensemble members have significant differences from each other. We believe that**

**our results highlight the necessity of the ensemble of different meteorological inputs for radionuclide dispersion modeling. Please see the response to l. 496.**

BRIOUDE, J., ARNOLD, D., STOHL, A., CASSIANI, M., MORTON, D., SEIBERT, P., ANGEVINE, W., EVAN, S., DINGWELL, A. & FAST, J. D. 2013. The Lagrangian particle dispersion model FLEXPART-WRF version 3.1. *Geoscientific Model Development,* 6**,** 1889-1904.

GALMARINI, S., BIANCONI, R., KLUG, W., MIKKELSEN, T., ADDIS, R., ANDRONOPOULOS, S., ASTRUP, P., BAKLANOV, A., BARTNIKI, J. & BARTZIS, J. 2004. Ensemble dispersion forecasting—Part I: concept, approach and indicators. *Atmospheric Environment,* 38**,** 4607-4617.

GRYTHE, H., KRISTIANSEN, N. I., GROOT ZWAAFTINK, C. D., ECKHARDT, S., STRÖM, J., TUNVED, P., KREJCI, R. & STOHL, A. 2017. A new aerosol wet removal scheme for the Lagrangian particle model FLEXPART v10. *Geoscientific Model Development,* 10**,** 1447-1466.

KARION, A., LAUVAUX, T., LOPEZ COTO, I., SWEENEY, C., MUELLER, K., GOURDJI, S., ANGEVINE, W., BARKLEY, Z., DENG, A. & ANDREWS, A. 2019. Intercomparison of atmospheric trace gas dispersion models: Barnett Shale case study. *Atmospheric Chemistry and Physics,* 19**,** 2561-2576.

MASSON, O., BAEZA, A., BIERINGER, J., BRUDECKI, K., BUCCI, S., CAPPAI, M., CARVALHO, F., CONNAN, O., COSMA, C. & DALHEIMER, A. 2011. Tracking of airborne radionuclides from the damaged Fukushima Dai-ichi nuclear reactors by European networks. *Environmental Science & Technology,* 45**,** 7670-7677.

PISSO, I., SOLLUM, E., GRYTHE, H., KRISTIANSEN, N. I., CASSIANI, M., ECKHARDT, S., ARNOLD, D., MORTON, D., THOMPSON, R. L. & GROOT ZWAAFTINK, C. D. 2019. The Lagrangian particle dispersion model FLEXPART version 10.4. *Geoscientific Model Development,* 12**,** 4955-4997.

STOEHLKER, U., NIKKINEN, M. & GHEDDOU, A. 2011. Detection of radionuclides emitted during the Fukushima nuclear accident with the CTBT radionuclide network. *Monitoring research review: Ground-based nuclear explosion monitoring technologies***,** 715-724.

STOHL, A., FORSTER, C., FRANK, A., SEIBERT, P. & WOTAWA, G. 2005. The Lagrangian particle dispersion model FLEXPART version 6.2. *Atmospheric Chemistry and Physics,* 5**,** 2461-2474.

TEN HOEVE, J. E. & JACOBSON, M. Z. 2012. Worldwide health effects of the Fukushima Daiichi nuclear accident. *Energy & Environmental Science,* 5**,** 8743-8757.

---

## Author Response (AR3)

Dear Editor,

Thank you for providing us with the additional reviews on our revised manuscript. We appreciate the feedback from the reviewers. We have addressed all of their concerns in our revised manuscript and have provided detailed responses to their comments below.

Thank you for your time and consideration.

**Referee #1:**

General comments:

1) The article is centered on the consequences of hypothetical Barakah powerplant accidents in Qatar only, and I'm wondering if the paper is within the scope of the ACP journal in its present form. The results of this paper made me wonder why focusing on Qatar and the Barakah powerplant in the first place, while the results are relevant for a larger region. The authors justify their analysis because new nuclear facilities are planned in the Middle East/North Africa region. Why not extending the analysis to the region covered by the WRF domain used in the paper for instance? Then, the Barakah power plant is not the only one in the region. The Bushehr powerplant in Iran, the second active nuclear power plant in the region if I'm not mistaken, could also be used in the FLEXPART simulations. This powerplant is located within the WRF simulation domain used by the authors, and it should not be too much of a problem to add the calculations from hypothetical accidents from this powerplant.

✓ **We would like to thank the reviewer for their valuable input regarding the inclusion of Bushehr nuclear power plant in the updated version. We have taken the suggestion into consideration, and we have implemented the emission from Bushehr NPP for a one-year period using the ERA5-WRF dataset, which has been fed into FLEXPART-WRF. In the subsection 3.4, we have included simulations of this sensitivity test, along with three other sensitivity runs that consider different release conditions, to also account for the points raised by Referee 2. These simulations are then compared to the control run.**
**Lines 79-84:**

"Moreover, we investigated the sensitivity of our simulations to different turbulence schemes under convective conditions, as well as variations in the vertical profile, temporal profile, and point of emission. Specifically, we examined two different vertical profiles, two points of emission, and two temporal profiles of emissions, with one main variant and one secondary variant for each variation. The main variants were used in most of the manuscript, while the secondary variants were used in a small sensitivity test to evaluate the impact of these variations on our results."

**Lines 229-232**:

"The third sensitivity test, designated as the Release Location Sensitivity Test (RLST), is conducted by releasing radionuclides from the Bushehr nuclear power plant (Bu-NPP) and investigating the impact of source-receptor position on the transport of radionuclides, particularly in relation to seasonal variations in atmospheric patterns."

**Lines 476-494:**

"The RLST run investigates the effect of changing the emission point from B-NPP to Bu-NPP. As expected, this change leads to significant differences in the amount of radioactive materials transported to Qatar. In general, the results show that the occurrence of a nuclear accident in Bu-NPP will be associated with the transfer of less radioactive materials to Qatar, which is the case almost throughout the year. The upper quartile of the full-year median of $^{131}$I concentrations and $^{137}$Cs deposition in RLST simulations a decrease by a factor of approximately 2.2 and more than 3, respectively. The seasonal variability of the transport and deposition of radioactive materials from Bu-NPP is a noteworthy aspect to consider in our analysis. the simulations of the RLST run show that $^{137}$Cs deposition is more prominent in winter and spring compared to other seasons. Moreover, the upper quartile value of $^{131}$I concentrations from Bu-NPP exceeds that of B-NPP by more than 80% in spring which is in contrast to the simulations of other seasons."

[Figure]

Figure 1 A:  These boxplots depict the **96-hour integrated simulations of** [131]I concentrations (**ng m^-3**) based on ERA5-WRF inputs. The simulations include control runs shown in cyan, and sensitivity runs with the skewed turbulence model (STM) in brown, modified release height from the 100-300m agl to the 100-700m agl (RHST) in blue, increased release duration from 24 to 96 hours (CRST) in pink, and a change in the release location from Barakah to Bushehr nuclear power plant (RLST) in green. B:  These boxplots show the same for total deposition of 137Cs (**ng m^-2**). The quartiles of the simulations are depicted by the box borders, while the whiskers illustrate the range between 1.5 times the interquartile range above the upper quartile and below the lower quartile.

**Lines 618-619:**

**"Changing the emission point from B-NPP to Bushehr-NPP leads to less transfer of radioactive materials to Qatar."**

2) Then the authors could just show the total deposition from both powerplants, with contours that would represent the contribution from Barakah or Bushehr within the WRF domain. That way, the authors would treat the potential of nuclear accidents within a larger region, and from the two active power plants of the region. In that case, the paper would better fit in the ACP scope. The authors could still present a zoom on the Qatar country.

✓ **The main objective of our study is to investigate the impact of meteorological inputs and simulation codes on the transport of particles, along with their seasonal and diurnal variations. We now also include the Bushehr-NPP as suggested by the referee, focusing on Qatar as the main catchment area for the impact of any single point of release independently. Further investigating multiple emission and receptor points in a wide range would indeed be a worthwhile topic for future studies.**

3) The validation of the met models could still be limited to the stations in Qatar. the validation of the met models against observations need to be improved. We don't know what kind of instruments generated those observations, and how the stations are covering Qatar. I doubt that the 140 stations have the same quality standards in terms of observation. The authors should select the stations with the best instruments, and for specific part of the domain (shore, inland, etc.).

✓ **We have incorporated information regarding the quality of the meteorological data used in the study into the revised text.**

**Lines 198-202:**

**"The GSOD data is subject to a rigorous two-tier quality control (QC) process (https://www.ncei.noaa.gov/data/global-summary-of-the-day/doc/readme.txt, last accessed: May 5, 2023). The first level entails the application of meticulous automated QC procedures that effectively purge the raw data of random errors. The second level of QC is performed to create daily summaries. However, there is still a slight likelihood of unknown errors being present within the GSOD dataset."**

✓ **We would like to emphasize that we utilized the majority of available meteorological observations to evaluate the inputs of atmospheric transport models at regional scales, which may also impact the quality of simulations over the catchment of interest. In addition, we did not find specific criteria or information for distinguishing the quality of stations in coastal areas or in any other regions. And we also did not find data susceptibilities in the used observations. While we acknowledge that there may be variability in the quality of meteorological data across the study area, we relied on the quality measurement techniques employed by the data producer, a widely recognized authority in the field.**

✓ **Regarding the comprison of the met models with the stations in Qatar,**

**Lines 270-276:**

**"In order to deepen our understanding of the model performance at the primary emission source and receptors in Qatar, we conducted a similar analysis on 12 stations within the area encompassing the B-NPP and the state of Qatar (S-5, -6, and -7). Although some variations are noted, such as an improvement in the correlations between simulated and observed wind speed, the overall patterns of performance for the input datasets over this region remain consistent with those identified earlier. Based on these findings, we infer that the FNL- and ERA5-WRF (downscaled) datasets exhibit a better correlation with observations. However, this comes at the expense of increased error and bias values when compared to the FNL and GFS datasets."**

4) The statistical analysis should be modified. For temperature, the authors could compare mean diurnal profiles for each season. For precipitation, the authors could present maps of

seasonal mean precipitation observed by the stations, and simulated by the models, instead of comparing the precipitation for each site.

✓ **Due to limited access to hourly-resolved data, we are unable to compare the diurnal distribution of temperature for each season, as stated in the previous version of the manuscript (line). To address the referee's comment, we produced seasonal boxplots for precipitation, wind speed, and temperature (available in supplementary S4 and S7). This analysis contributes to the discussion of the potential underestimation or overestimation of these meteorological factors in our simulations.**

5) Figure 2, 12 and 13 are hard to interpret. The scatterplots are impossible to read (the authors should use a pdf or something else).

✓ **Done. The quality of these images has been improved.**

6) I'm wondering if those plots should be directly in the supplement materials.

✓ **We have moved Figure 2, along with other figures related to the evaluation of meteorological inputs, to the supplementary (S2-7). Figures 12 and 13 (Figures 11 and 12 in the new version) were retained in the main body due to their importance in the discussion.**

7) Figure 5 is hardly used in the paper, and could be put directly in the supplement.

✓ **This figure is necessary for discussing the relationship between particle ages and the deposition of Cs137 (deposition speed) in the study area. Hence, we have opted to retain Figure 4 (previously Figure 5) in the revised manuscript.**

**Reviewer 2:**

EMISSION SCENARIO

1) The selection of the emission scenario limits the applicability of the study. Taking an envelope from Fukushima accident (different reactor type and thermal power) and also constant emission for 24 hours at a constant height is too limited to draw sound conclusions and, specially, to be used in the formulation of preparedness plans. It is clear that the emission height and the shape of the release (some hours with more release, others smaller, others at higher levels etc) significantly affects the outcome of the simulations and the transport and deposition patterns. I would suggest that there is a preliminary work to try to identify those realistic scenarios that provide the right starting point. There is no need to have many of such scenarios but a worst case (low probability high impact) and a more

probable (more probable with less impact) should be approached. In this way the starting point of all the study would be more realistic and sustain better the conclusions.

✓ **We would like to thank the referee for the suggestion. While the analysis of release conditions was not the focus of our study, this point is valuable in improving our work. In light of the computational load required, we conducted three sensitivity runs using ERA5-WRF (with FLEXPART-WRF), examining the impact of changes in height, duration, and point of emissions on the simulations. These results are discussed in a new subsection 3.4. We note that the investigation of other possible release conditions could be explored further in a separate study. More details of these changes are presented below**

**Lines 79-84: "Moreover, we investigated the sensitivity of our simulations to different turbulence schemes under convective conditions, while considering the effect of variations in the vertical profile, temporal profile, and point of emission. Specifically, we examined two different vertical profiles, two points of emission, and two temporal profiles of emissions, with one main variant and one secondary variant for each variation. The main variants were used in most of the manuscript, while the secondary variants were used in a small sensitivity test to evaluate the impact of these variations on our results."**

**Lines 446-484:**

[revised manuscript text omitted]

**Referee #2:**

DOSIMETRIC DISCUSSION

2) In order to talk about relative contributions, there is no need to go towards the actual dose calculations and could be worked out with the concentrations and depositions. The moment the doses are provided the consequences are thought and this is a very sensitive issue. Should a dosimetric approach be desired, then it is important to have an improved emission scenario, as explained in the former section. This should be also followed by a better description of the approximations taken to calculate doses (semi-infinite loud, dosimetric dose coefficients etc) and compare them with the regulatory thresholds. In this way, the work would be indeed valuable for preparedness plans and understanding and definition of potential mitigation measures preparations according to regions and seasons.

✓ **Indeed, the scope and the primary focus of our study was not on dosimetry, but we received a request from one of the reviewers in a previous round to pay more attention to this topic in our analysis. To avoid any unnecessary sensitivity regarding our results, we now present the simulations in the form of concentrations (ng/m3) and deposition (ng/m2) in the revised version, as suggested. Additionally, we conducted three sensitivity runs to investigate the effects of changes in emission scenarios on our simulations, independent of dosimetry considerations.**

In addition, please consider the following remarks:

3) The work presents and ensemble of 4 members two of which are of similar nature. Therefore it is hard to argue this is a real ensemble. Please be so kind to mention it as a mini-ensemble.

✓ **Done. All occurrences of ensemble in the manuscript changed to mini-ensemble.**

4) Why was not ERA5 used directly to drive the simulations? Why was not the ECMWF forecasted fields used to drive the simulations?

✓ **We have now added consideration and discussion for the reason we did not use ERA5 directly for FLEXPART:**

**Lines 183-188:**

**Though we devoted efforts to incorporate ERA5 reanalysis data directly as inputs for FLEXPART, we were hindered by technical difficulties encountered with the data preprocessor tool, flex_extract, thereby impeding our ability to do so. While we were working to resolve the technical issues with the developers, a solution was not readily available during the writing and revision of this manuscript. We have indirectly employed ERA5 reanalysis data from the Research Data Archive (RDA) by incorporating it into the WRF model and subsequently into the FLEXPART-WRF.**

✓ **We refer the referee/reader to an older communication of the primary author with the FLEXPART developers regarding the problem with the retrieval of ERA5 using Flex_extract. The details of our communication can be found in the following link: https://www.flexpart.eu/ticket/300.**

5) The former two questions relate to the potential applicability of the study: do we want a tool and system to prepare or do we want a system to respond? if the second, then the computational speed is key and one could argue about the need of having the downscaling approach. It would be good to have further discussion on this.

✓ **We see the scope of our study as creating awareness and preparing for a potential nuclear accident, rather than responding to a real occurrence. Specifically, we aimed to identify areas and time periods with a higher risk of radionuclide deposition/concentrations and to guide preparedness, mitigation, and adaptation plans. Our results can also assist in selecting suitable locations for setting up radioactivity measurement stations to monitor any unusual changes in the values of concentrations and settlements of radioactive materials. Additionally, the simulations of the member based of GFS forecast can still provide insights for an ongoing nuclear accident. We agree that the degree of uncertainty associated with these results is likely to be higher compared to the output ensembles (lines 86-90).**

6) It would be interesting to see whether a higher resolved WRF would make a difference.

✓ **In an attempt to comply with one of the editor's suggestions in the previous round of revision, we made an effort to utilize FNL inputs downscaled to 4 and 2 km resolutions. Despite our best efforts, we encountered technical difficulties that**

**hindered our ability to reconcile the simulation output with the observations. Consequently, we made a decision to omit the 4km and 2km sensitivity analysis from this paper and defer it to future research.**

7) Figures 6,7,8 (specially 6) have a color scale hard to read and to differentiate the isolines.

✓ **Done. In order to enhance the visual clarity and facilitate interpretation of the figures, we have incorporated a hash pattern into them. As an example, please refer to the figure below:**

[Figure]

✓
✓ **Figure 3 Seasonal median of 96-hour integrated $^{131}$I concentrations ($^{131}\text{I}^{\text{intg\_conc\_seas}}$). The contour lines (in local time, hours of the day) depict the seasonal median of the Lagrangian particle release time coinciding with the maximum $^{131}\text{I}^{\text{intg\_conc\_seas}}$ found in each 96-hour run.**

8) If the aim is to prepare the tools, materials and approaches for the preparedness plans, it would be interesting to see whether other NPPs in the region (the one operating or planned to be operative in the future) are

   ✓ **We have followed up on this comment, also considering a similar request by the first reviewer. We have now added an investigation of the concentrations and deposition of radioactive materials from the Bushehr nuclear power plant in the region. For further details, we refer to our response to Reviewer 1.**

MORINO, Y., OHARA, T. & NISHIZAWA, M. 2011. Atmospheric behavior, deposition, and budget of radioactive materials from the Fukushima Daiichi nuclear power plant in March 2011. *Geophysical research letters,* 38.
PISSO, I., SOLLUM, E., GRYTHE, H., KRISTIANSEN, N. I., CASSIANI, M., ECKHARDT, S., ARNOLD, D., MORTON, D., THOMPSON, R. L. & GROOT ZWAAFTINK, C. D. 2019. The Lagrangian particle dispersion model FLEXPART version 10.4. *Geoscientific Model Development,* 12**,** 4955-4997.

---

## Author Response (AR4)

Dear Editor,

We would like to thank you for your valuable contribution, which has significantly improved the quality and clarity of our manuscript. We have addressed all comments and provide detailed responses below:

1) Bu-NPP: In the abstract and conclusion, and to some extent in the main article, you don't mention the results from the Bu-NPP. You showed that the contribution is in fact the highest in springtime. You don't show any map of iodine or Cesium concentrations from the Bu-NPP in the paper. However, I assume the transport of radioactive materials pass over the northern part of Qatar and not the south. If the map is significantly different than the runs with B-NPP, you should add a map of the results showed on figure 10A and develop a little bit the results around Bu-NPP.

   ✓ **We have carefully considered your feedback and made the necessary revisions accordingly. In response to your suggestion, we have now incorporated a discussion and included maps pertaining to the emission from the Bushehr NPP in the abstract, conclusion, and relevant sections of the main article. Specifically, we have included a map in Figure 11, displaying iodine and Cesium concentrations from the Bu-NPP, thus providing a visual representation of the results.**

**Lines 27-30:**

**"As part of a sensitivity analysis involving different model setups, changing the emission point from B-NPP to Bushehr-NPP (Bu-NPP) results in a reduced transfer of radioactive materials to Qatar, except in the spring season. Bu-NPP simulations reveal distinct spatial patterns, with peak $^{131}$I concentrations and $^{137}$Cs deposition observed in the northern and eastern Qatar during winter and spring."**

**Lines 51-52:**

**"The Barakah Nuclear Power Plant (B-NPP) is the latest example, following the Bushehr NPP (Bu-NPP), to become operational in a region with unique climatological conditions."**

**Lines 489-497:**

**"In Figure 11-A, a distinct spatial pattern emerges in the Bu-NPP simulations when compared to the B-NPP simulations (Figs. 5, 6, and 7). The simulated $^{131}$I$^{intg\_conc\_full}$ exhibit relatively high values in the northern and eastern regions, including Doha, particularly during the winter and spring seasons. This occurrence can be attributed to the downward movement of westerlies, facilitating the transport of air masses**

enriched with $^{131}$I towards these areas. The full-year median of the total $^{137}$Cs deposition ($^{137}$Cs$^{tot\_depos\_full}$) (Fig. 11-B) closely follows this temporal pattern, showing prominent peaks during the spring and winter seasons within the northern and eastern regions of Qatar. Conversely, during the fall and summer seasons, the levels of $^{131}$I$^{intg\_conc\_full}$ and $^{137}$Cs$^{tot\_depos\_full}$ are greatly reduced. The age of particles corresponding to high levels of $^{131}$I and $^{137}$Cs simulations indicates their earlier entry into the northern half of Qatar during these two seasons, reaching the region within approximately 40-50 hours and 60 hours after release, respectively."

**Lines 506-509**:

[Figure]

Figure 1:  Simulated $^{131}$I$^{intg\_conc\_full}$ (A) and $^{137}$Cs$^{tot\_depos\_full}$ (B) based on ERA5-WRF inputs originating from Bushehr NPP. The contour lines are the full-year median of age spectra coinciding with the maximum $^{131}$I concentrations and the completion of $^{137}$Cs deposition found in each 96-hour run.

**Lines 633-639**:

"As anticipated, due to the longer distance, the change in the emission point from B-NPP to Bushehr-NPP (Bu-NPP) results in reduced transfer of radioactive materials to Qatar, except during the spring season. The simulations from Bu-NPP exhibit distinct spatial patterns compared to those from B-NPP. Concentrations of $^{131}$I peak in the northern and eastern regions during winter and spring, which can be attributed

to a southward shift of the westerlies. The deposition of $^{137}$Cs follows a similar pattern. The entry of particles inducing high intensities of $^{131}$I and $^{137}$Cs over the northern region of Qatar occurs within approximately 40-50 hours and 60 after their release, respectively."

2) Line 50: You should also mention that the Bu-NPP is the second power plant in the MENA region.

✓ **Done. Lines 51-52 "The Barakah Nuclear Power Plant (B-NPP) is the latest example, following the Bushehr NPP (Bu-NPP), to become operational in a region with unique climatological conditions."**

3) Section 3.2: The results in section 3.2 would be more easily accessible if they were listed in a table.

✓ **After careful consideration, we have found that it is best to retain the current presentation style in Section 3.2 for conveying information regarding particle ages and their distribution. But, we have enhanced the presentation of results in subsection 3.1. While we have decided to retain the figures S2 and S3 showcasing scatter plots of the modeled and observed meteorological fields, we have also incorporated Table 3 specifically addressing the statistics of model performance against observations. We note that the table focuses on the statistics related to observations, and does not include model intercomparisons.**

**Lines 289:**

Table 3: Comparison of modelled and observed daily precipitation, wind speed, and temperature.

| Variable | Mini-ensemble members | Spearman Correlation coefficient | RMSE | MBE |
|---|---|---|---|---|
| Precipitation (mm) | GFS | 0.42 | 3.1 | -0.18 |
| | FNL | 0.42 | 3 | -0.18 |
| | FNL-WRF | 0.37 | 3.4 | -0.27 |
| | ERA5-WRF | 0.44 | 3.3 | -0.24 |
| mean | | 0.41 | 3.2 | -0.22 |
| Wind Speed (m/s) | GFS | 0.58 | 1.7 | -0.13 |
| | FNL | 0.58 | 1.7 | -0.15 |
| | FNL-WRF | 0.64 | 4.1 | 0.62 |
| | ERA5-WRF | 0.65 | 1.5 | 0.51 |
| mean | | 0.61 | 2.2 | 0.21 |
| Temperature (k) | GFS | 0.97 | 2.1 | 0.02 |

| | | FNL | 0.98 | 2 | 0.02 |
|---|---|---|---|---|---|
| | | FNL-WRF | 0.98 | 1.8 | 0.07 |
| | | ERA5-WRF | 0.97 | 2 | -0.33 |
| | mean | | 0.975 | 1.975 | -0.055 |

4) equation 3, line 116: either you explain each member of the equation, or you don't show it.

   ✓ **Done. We have addressed this by providing explanations for each member of the equation.**

**Lines 112-118:**

**"The minimum value of time step $\Delta t_i$ is 1 second. $\Delta t_i$ is used only for the horizontal turbulent wind components of Equation 3.**

$$\Delta t_i = \frac{1}{ctl} min(\Delta\tau_{L_\omega}, \frac{h}{2\omega}, \frac{0.5 \frac{\partial\sigma_\omega}{}}{\frac{\partial\sigma_\omega}{\partial z}}) \ (3)$$

**In Equation 3, $h$ represents the height of the atmospheric boundary layer and $z$ denotes the height of the model level. The constant $ctl$ represents a predefined characteristic time scale. $\tau_{L_\omega}$ is the Lagrangian timescale for the vertical velocity autocorrelation. Additionally, $\omega$ represents the turbulent vertical wind component, while $\sigma_\omega$ represents its standard deviation. To solve the Langevin equation for the vertical wind component, a shorter time step $\Delta t_\omega = \frac{\Delta t_i}{ifine}$ is used."**

5) Line 267-271: If you reach a resolution of 1km, you can for instance switch off a convective scheme that would be designed for coarser resolution because most of the deep convection will be resolved. So going to a finer scale might help in reducing the number of subgrid phenomena that need to be simulated.

   ✓ **In response to your suggestion, we have incorporated the following lines into the main manuscript:**

**Lines 264-267:**

**"While this study did not specifically investigate model resolutions finer than 10 km, it is important to note that increasing the model resolution to 1 km or less provides the capability to resolve finer subgrid phenomena. This allows for the deactivation of convective schemes designed for coarser resolutions, as the resolved deep convection becomes more prominent."**